# Sufficient conditions for rapid range expansion of a boreal conifer

Roman J. Dial[1✉], Colin T. Maher[2✉], Rebecca E. Hewitt[3,4✉] & Patrick F. Sullivan[2✉]

Unprecedented modern rates of warming are expected to advance boreal forest into Arctic tundra[1], thereby reducing albedo[2–4], altering carbon cycling[4] and further changing climate[1–4], yet the patterns and processes of this biome shift remain unclear[5]. Climate warming, required for previous boreal advances[6–17], is not sufficient by itself for modern range expansion of conifers forming forest–tundra ecotones[5,12–15,17–20]. No high-latitude population of conifers, the dominant North American Arctic treeline taxon, has previously been documented[5] advancing at rates following the last glacial maximum (LGM)[6–8]. Here we describe a population of white spruce (*Picea glauca*) advancing at post-LGM rates[7] across an Arctic basin distant from established treelines and provide evidence of mechanisms sustaining the advance. The population doubles each decade, with exponential radial growth in the main stems of individual trees correlating positively with July air temperature. Lateral branches in adults and terminal leaders in large juveniles grow almost twice as fast as those at established treelines. We conclude that surpassing temperature thresholds[1,6–17], together with winter winds facilitating long-distance dispersal, deeper snowpack and increased soil nutrient availability promoting recruitment and growth, provides sufficient conditions for boreal forest advance. These observations enable forecast modelling with important insights into the environmental conditions converting tundra into forest.

In contrast to expected range expansions[1], the primary response of North American spruce populations to recent warming at the Arctic forest–tundra ecotone has been a growth-form shift from stunted 'krummholz' to upright trees[9–13], an increase in stand density[11,14–21] or both[10–14]. However, even advancing boreal conifers[12,15,21] cannot keep pace with ongoing isotherm movement[5]. The observed rates of treeline advance[5] challenge[5,17,18,20] simulation models[1,22–24] to reformulate forecasts of boreal migration in response to climate change. To match post-last glacial maximum (LGM) migration rates[7] of 3–4 km per decade, modern range expansion by white spruce requires long-distance seed dispersal (LDD)[25], successful germination by cold-sensitive seeds[19], rapid growth under limiting conditions[17] and early sexual reproduction[13].

White spruce dispersal distances are generally <100 m (refs. [26,27]), with successful germination requiring growing season temperatures of ≥10 °C (refs. [13,19]). Because seed production in quantity begins at 30 years of age[26], an estimate[25] of range expansion is 0.03 km per decade, two orders of magnitude slower than paleo-rates[7]. Detection of range expansion through remote sensing[5,18,20] often relies on repeat growing season imagery of treelines visible by virtue of tree height and density ('established treelines'). However, young, sparsely distributed colonists extending a species range may go undetected without extensive field-based surveys.

We describe a large, expanding population of young, vigorous, sexually reproducing spruce, thriving within an Arctic watershed previously unoccupied by spruce for millennia[28] and advancing at rates approaching post-LGM migration out of glacial refugia[7]. Using satellite imagery (Fig. 1 and Supplementary Figs. 1–13) and field campaigns (Supplementary Fig. 14), we document white spruce dispersal over Alaska's Brooks Range, a 1,000-km Arctic mountain range (Fig. 1a) long considered a barrier to forest advance[22]. The oldest trees appear to have colonized during the late nineteenth and early twentieth centuries by dispersing over a mountainous divide from established treelines in a basin supporting spruce for 6,000 years[29]. Our observations suggest that winter winds, deep snow and greater nutrient availability provide for rapid individual and exponential population growth, propelling the population northwards at >4 km per decade, faster than for all modern conifer treelines previously measured[5]. These environmental factors are associated with rising temperatures and decreasing sea ice[30–32], highlighting interconnections between marine and terrestrial components in a rapidly changing Arctic. Identifying conditions for boreal forest advance will help parameterize and validate simulations aiming to forecast biome shifts.

## Forest–tundra ecotone advance

During a field survey in 2019, we discovered a previously unknown population of white spruce in the largest tributary watershed of the Noatak River basin. The Noatak drains the western Brooks Range and at

[1]Institute of Culture and Environment, Alaska Pacific University, Anchorage, AK, USA. [2]Environment and Natural Resources Institute, University of Alaska Anchorage, Anchorage, AK, USA. [3]Department of Environmental Studies, Amherst College, Amherst, MA, USA. [4]Center for Ecosystem Science and Society, Northern Arizona University, Flagstaff, AZ, USA. ✉e-mail: roman@alaskapacific.edu; cmbristlecone@gmail.com; rhewitt@amherst.edu; pfsullivan@alaska.edu

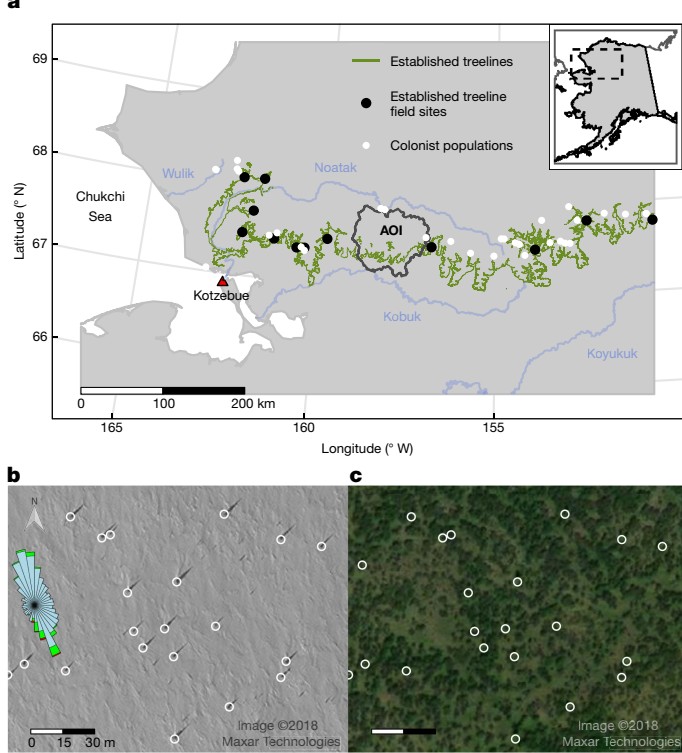

**Fig. 1 | Winter satellite scene shows spruce undetectable on summer scene.**
**a**, Northwest Alaska, USA, west of 150.5° W showing established treelines
separating boreal forests from Arctic tundra in Alaska's Brooks Range as an
olive green line. Black circles indicate established treeline study sites, and
white circles show additional known colonist populations >1 km from
established treelines. The black outline indicates the AOI enclosing four
watersheds in the Arctic Noatak basin and four watersheds in the boreal Kobuk
basin. Blue lines correspond to rivers mentioned in the text. The red triangle
indicates the location of Kotzebue. **b**, Snow-covered satellite scene (0.5-m
resolution; WorldView-1 panchromatic ©2018 Maxar Technologies, 26 March
2018) showing 24 trees casting shadows as digitized using Google Earth Pro
(GEP) super-overlays of the WorldView-1 imagery. Super-overlays degrade
imagery, and smaller tree shadows were therefore missed by the digitizing
technician (Supplementary Information sections 1.2 and 1.3). Approximately
6,000 such shadows were digitized within the Noatak watersheds of the AOI
polygon in **a**. Their densities are shown in Extended Data Fig. 1b, and their
locations are indicated in Supplementary Fig. 1. Snow drifts and the wind rose
(data from a remote automated weather station (RAWS) 14 km west of the
image location) indicate strong southerly winds. **c**, Same scene as in **a** but
during the growing season (0.5-m resolution; Vivid ©2018 Maxar Technologies,
7 July 2018) showing *Salix* shrubs ≤2.5 m tall (dark green). Both scenes enclose
24 digitized *Picea* trees ≥3 m tall with bases marked by white circles. Scenes are
located near 67.56° N, 158.09° W at the southwest corner of the rectangle
labelled 'Simulated population area' in the centre of Extended Data Fig. 1a and
in red rectangles in Supplementary Figs. 1–4.

Kotzebue empties into the Chukchi Sea (Fig. 1a), which is experiencing
the fastest rate of sea-ice decline in the Arctic Ocean[32]. Across $10^3$ km²
of Arctic tundra within the area of interest (AOI; Fig. 1a and Extended
Data Fig. 1a), we geolocated 6,758 white spruce trees (Supplementary
Information section 1, Extended Data Fig. 1b–d and Supplementary
Figs. 1–3) using very high-resolution (0.5-m) snow-covered panchro-
matic scenes (Fig. 1b and Supplementary Figs. 1–13) and >400 km of
field transects (Supplementary Fig. 14) where we measured the ecological
attributes of individual trees.

Juvenile members of the colonist population were separated from
established treelines of the adjacent forested Kobuk River basin by
up to 42 km (Extended Data Fig. 1d). In the field, nearly all individuals

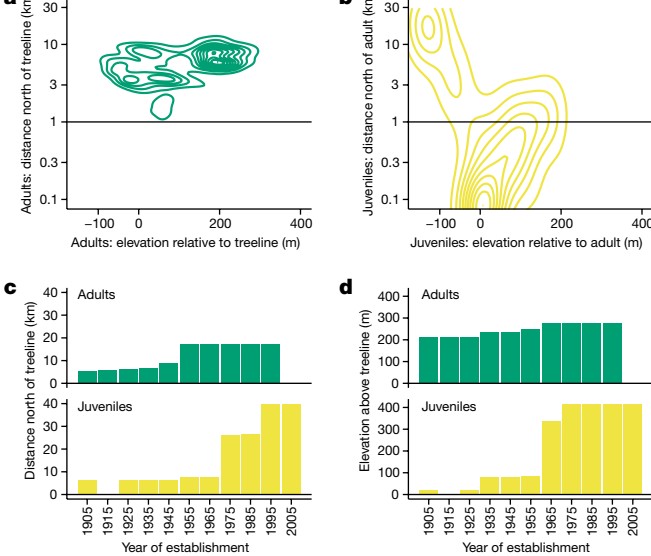

**Fig. 2 | Juveniles have dispersed far from adults while adults have dispersed
far from treelines. a**, Bivariate dispersal kernel ($n = 5,986$) as a two-dimensional
density plot giving adult colonist distances north of and elevations relative to
established treelines in the Kobuk basin. **b**, Bivariate dispersal kernel ($n = 770$)
giving juvenile colonist distances north of and elevations relative to adult
colonists. **c**, Furthest-forward colonists stratified by 10-year class for date of
establishment showing northward distances from established treelines for
adults (green, $n = 131$) and juveniles (yellow, $n = 311$). **d**, Furthest-forward
colonists stratified by 10-year class for date of establishment showing elevation
above established treelines for adults (green, $n = 131$) and juveniles (yellow,
$n = 311$). Supplementary Information section 1 presents calculations.

appeared <100 years old and were growing rapidly (see 'Population and
individual growth'). Trees of all sizes (≤11 m) displayed erect, healthy,
symmetric crowns and long leaders. We observed a near absence of tree
islands[13], krummholz, layering or other asexual growth typical at estab-
lished treelines[8–20]. During a mast event in 2020, cone crops on some
trees exceeded $10^3$ cones, with 99% of cone-bearing colonists ≥2.5 m
tall (Supplementary Information section 1.6). The eight oldest colonists
were established in 1901–1933 and grew near the Kobuk–Noatak divide,
5.8–7.0 km from an established treeline (Fig. 2c, Extended Data Fig. 1c
and Supplementary Fig. 2), suggesting several founding LDD events.

## Patterns of expansion

Trees ≥2.5 m tall ('adults') grew considerably higher in elevation and/or
further north than the nearest established treelines (Fig. 2a, Extended
Data Fig. 1c and Supplementary Figs. 1, 2 and 4). Individuals <2.5 m tall
('juveniles') were often higher in elevation and/or further north than
adults (Fig. 2b). Whereas adults had reached a maximum distance from
established treelines by the mid-twentieth century, juveniles have
continued to move northwards and upwards during the twenty-first
century (Fig. 2c,d). For 5,988 mapped adults, the maximum northward
displacement from the nearest established treeline was 0.16° (17.4 km
north; 95% quantile , 9.7 km north; Fig. 2a and Supplementary Infor-
mation section 1). The migration rate of the furthest-forward adult
(1.4 km north per decade) still substantially lags the accelerating rate of
isotherm advance[5], but far surpasses the migration rate (0.005 km per
decade) estimated[21] for treelines in the lower Noatak basin 175 km west.

A full depiction of range expansion includes a dispersal kernel[25],
portrayed here as a bivariate distribution of juvenile distance from
and elevation relative to the nearest adult ($n = 770$ juveniles; Fig. 2b).
Mapped juveniles were mostly near adults (median distance, 59 m).
However, at five locations ≥1.2 km from one another, 14 juveniles

were 9.3–22.9 km north of the nearest adult and 26.6–40.2 km north (0.24–0.36°) of the nearest established treelines (Fig. 2c, Extended Data Fig. 1d and Supplementary Fig. 3). These distant juveniles were each <35 years old (estimated establishment year of 1989–2004; Supplementary Information sections 2 and 3), growing vigorously (height added from 2015–2020: mean, 52%; s.d., 13%; $n$ = 8), without evidence of krummholz, layering or other asexual reproduction. Some appeared mechanically broken through antler raking by migrating caribou, but most had upright leaders. We documented with global navigation satellite systems (GNSS) hundreds of juveniles up to 392 m above and 215 m below the elevation of the nearest adult (Supplementary Information section 1). Juveniles growing >100 m above the nearest adults grew as populations on ridgelines in low or dwarf shrub communities. Juveniles growing substantially below the nearest adults grew on river bars with tall willows or in tussock tundra.

Conventional white spruce silvics identify 45–60 m as the typical dispersal distance[26], closely bounding the median distance between juveniles and their nearest adults, and >300 m as LDD. Here 32% of juveniles were >300 m from the nearest adult and 13% were >800 m distant. Snow drifts visible on imagery and wind direction obtained from nearby weather stations show strong, frequent southerly winter winds (Fig. 1b), supporting hypotheses of frequent winter wind-driven dispersal from both empirical studies of modern dispersal[27,33] and a conceptual model of range expansion in white spruce following the LGM[6].

## Population and individual growth

A hallmark of plant invasion ecology is the recursive founding of small populations following LDD[25]. Applying this theory of invasion to white spruce in the upper Noatak basin implies that exponential population growth drives local range expansion forwards. To test this hypothesis, we reconstructed past populations in a well-sampled sub-watershed (rectangle in Extended Data Fig. 1a, detail in Extended Data Fig. 2a and Supplementary Figs. 1–4) with 1,000 resamples from establishment year probability distributions conditional on five height classes (Extended Data Fig. 2b). Fitting each simulation (Supplementary Information section 4.4) as population size $N(t) = N_0 e^{k(t-1900)}$, where $t$ is the year and $k$ is the exponential growth rate, gave a mean $k$ of 0.07 per year. On average, the simulated populations doubled each decade (median, 9.5 years; interquartile range, 8.7–10.6 years; Supplementary Information section 4.6) from 1900 to 1980 (Fig. 3a), a result that was robust to reducing the number of height classes to three (median, 9.8 years; interquartile range, 8.8–10.8 years; Supplementary Information section 4.8). An example simulation (Supplementary Video 1) shows the spatial clustering and rapid population growth during the 1970s.

Individual growth is an implicit constraint of vital rates in populations. We measured individual growth with three metrics relative to Kotzebue July air temperature. The growth metrics included current annual lateral branch growth (CAG; Supplementary Information section 5) in adults during the warmest July (2019) of the continuous Kotzebue instrumental record (1937–2020); relative height growth rate (RGR; Supplementary Information section 6) in juveniles over the warmest five consecutive Julys (2015–2020); and main-stem radial growth (Supplementary Information section 7) over the warmest three consecutive decades (1989–2019) as compared with prior growth. Radial growth measures included raw ring width (RRW), basal area increment (BAI) and autoregressive residuals (ARs). The AR index reflects interannual variation with temporal autocorrelation removed ('pre-whitening', found here with best fit order of one; Supplementary Information section 7.2), whereas RRW and BAI reflect growth trends. Kotzebue July air temperature is probably very well correlated with July air temperature in the AOI, given the strong, consistent July lapse rates in the Baird Mountains (Extended Data Fig. 3 and Supplementary Information section 7.2).

All three growth metrics showed that colonists grew more rapidly than individuals at established treelines. Adult colonist CAG was

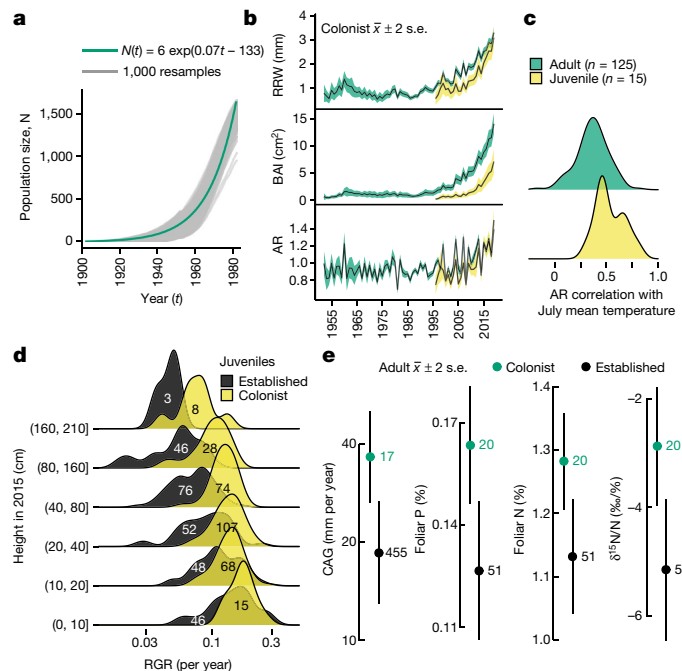

**Fig. 3 | Colonists grow faster than individuals at treelines. a,** Population exponential growth from 1900 to 1980. Monte Carlo simulations ($n$ = 1,000) of colonists in the central rectangle in Extended Data Fig. 1a are shown as grey lines. The green exponential curve corresponds to the mean of 1,000 runs. Details in Supplementary Information section 4. **b,** Colonist tree-ring (Extended Data Fig. 4) chronologies for adults (≥30 years, green, $n$ = 125) and juveniles (<30 years, yellow, $n$ = 15) determined with RRW, BAI and AR shown as mean ± 2 standard error (s.e.). Chronologies support five or more series. Details in Supplementary Information sections 7.1–7.7. **c,** Distributions (Gaussian kernel) of Pearson's correlation between AR and 1989–2019 July Kotzebue temperature for adult (green, $n$ = 125) and juvenile (yellow, $n$ = 15) colonists. Statistics in Supplementary Information sections 7.8–7.11. **d,** Juvenile RGR of colonists (yellow) and individuals at established treelines (black) by size class. Colonist sample sizes are in black text ($n$ = 300) while those for individuals at established treelines are in white text ($n$ = 271). Wald test $t$ = 4.46 for population–height interaction in 2015 using linear mixed-effects models; $m$ = 24 sites as random effect. Statistics in Supplementary Information section 6. **e,** Adult CAG and foliar P and N concentrations for colonists (green) and individuals at established treelines (black). Colonist sample size is in green while that for individuals at established treelines is in black. Circles with error bars give the mean ± 2 s.e. Wald tests for fixed effect: CAG, $t$ = 3.69 ($n$ = 17 colonists and 455 individuals at established treelines); P, $t$ = 3.62; N, $t$ = 3.34; $\delta^{15}$N/N, $t$ = 3.50 ($n$ = 20 colonists and 51 individuals at established treelines). $m$ = 18 levels for watershed as random factor. Statistics in Supplementary Information section 5. Because of the unbalanced design in all linear mixed models, $P$ values were not calculated.

almost double that at treelines (36 versus 19 mm per year; Fig. 3e and Supplementary Information section 5.1). Juvenile colonist RGR was greater than at established treelines with a difference that increased with height, such that RGR was 1.9 times that at established treelines in the height class of 80–160 cm (Fig. 3d and Supplementary Information section 6.2). Both radial growth indices of colonists increased exponentially from 1989 to 2019 (Fig. 3b and Extended Data Fig. 4). The exponential growth rates of adults and juveniles during the most recent decades contrast with earlier rates reaching back to the 1950s, including in comparisons of growth rates of all trees as juveniles (Extended Data Fig. 5 and Supplementary Information sections 7.9 and 7.10).

Correlations of both AR (Fig. 3c) and ln(BAI) (Supplementary Information section 7.7) with 1989–2019 July temperature were positive for all trees <30 years old and for 98% of older trees. To our knowledge, this is the highest proportion of adults responding positively

to temperature recorded among treeline sites in the Brooks Range[34,35]. However, the spatially comprehensive chronologies for the Brooks Range presented in refs. [34,35] end with the twentieth century, leaving uncertain how widespread recent rapid radial growth might be. A localized Brooks Range tree-ring chronology ending in 2011 showed a strong response to recent warming[36]. With sampling well below the treeline in the lower Noatak basin on a riverside terrace with relatively high soil nutrient availability in warm soils, this chronology contrasts with chronologies from nearby sites with cooler, more nutrient-poor soils that fail to show a similar response to warming[36].

## Environmental conditions

Rapid Arctic warming affects colonists through multiple pathways, including increased snowfall and improved nutrient availability, which probably interact to improve juvenile survival and adult growth[36,37]. To explain the longitudinal gradient in radial growth response to temperature across Alaska's Arctic treelines[34,35], Sullivan et al.[36] advanced the hypothesis of nutrient limitation induced by cold soils, whereby soil nutrient availability, access or both constrain growth and reproduction of individual trees[37]. In line with this hypothesis, we found higher foliar N and P concentrations and a higher ratio of $\delta^{15}$N to N (an index of mycorrhizal associations[38], where $\delta^{15}N = 10^3([(^{15}N/^{14}N)_{sample}/(^{15}N/^{14}N)_{standard}] - 1))$ in colonists (Supplementary Information section 5.2) than in established treeline populations, although colonist sample size was much lower (Fig. 3e). Greater access to nutrients increases growth[36,37] and may reduce mycorrhizal carbon costs.

Regional winter precipitation is increasing as a consequence of rapid regional warming (2.3 °C per century; Extended Data Fig. 6c) and sea-ice loss (Supplementary Information section 8)[30–32]. From 1979 to 2019, both the extent of open water in October in the Chukchi Sea (Extended Data Fig. 6b) and winter precipitation in Kotzebue (Extended Data Fig. 6a) increased. Modelled[30] and empirical[31,32] data suggest that increased open water in the Arctic Ocean during autumn leads to more wind[32] and deeper snow[30,31]. Increased wind over a sufficiently protective snowpack[9–13] facilitates the physical transport of seeds in LDD[27,33]. These conditions encourage colonization of tundra by boreal conifers well beyond current population boundaries, as predicted by invasion models[25].

Arctic winter precipitation offers a proxy for snow depth, a factor providing thermal insulation and promoting overwinter activity by soil microbes, thereby increasing soil nutrient availability during the subsequent growing season, an effect first proposed for Arctic shrubs[39,40]. Increases in winter and growing season soil temperatures are expected to increase nutrient availability, a known limiting factor for spruce seedlings in tundra[37]. Snow protects juveniles[41–43], and snowmelt reduces moisture limitation during the growing season[43]. Snowpack affects population-level responses to damaging winter winds because growth form determines seed production[9–13,17]. Taken together, these factors highlight the complex and nuanced interactions among nutrients, climate, vegetative growth and reproduction[17,36–43].

To investigate differences in colonist and established treeline climates, we compared simple bivariate climate envelopes using gridded climate data for 30-year means of July temperature and November–March precipitation. We chose July temperature because it was strongly correlated ($r = 0.99$, $n = 6,373$ sample locations) with growing season degree day sum (the sum of daily mean air temperatures across all days warmer than 4 °C), the top univariate predictor of white spruce presence at the treeline in the Brooks Range[44].

Both populations occupied areas with July mean air temperatures ≥10 °C (Fig. 4a,b), but climate envelopes were cooler and snowier for colonists (Fig. 4b) than for individuals at established treelines (Fig. 4a). The mode for established treelines was near that for Kobuk watersheds (Fig. 4c), whereas colonists occupied the snowier end of climate space in Noatak watersheds (Fig. 4d). The misalignment between the climate

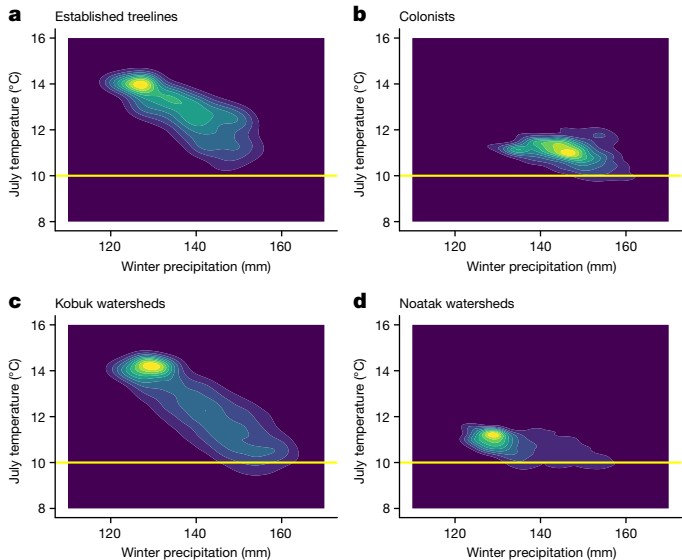

**Fig. 4 | Colonists occupy cooler, snowier climate space than individuals at treelines. a–d**, Climate envelopes for the AOI (Fig. 1a and Extended Data Fig. 1a) as 30-year means (1980–2010) extracted from 1-km gridded climate data and displayed as bivariate Gaussian kernel density plots (details in Supplementary Information sections 8.1 and 8.2). **a**, Established treelines in Kobuk watersheds adjacent to the Noatak watersheds containing colonists. **b**, Colonist populations in Noatak watersheds. **c**, Entirety of the four Kobuk watersheds in the AOI. **d**, Entirety of the four Noatak watersheds in the AOI. The established treeline climate envelope (**a**) is a subset of Kobuk watersheds (**c**), while the colonist population climate envelope (**b**) is a subset of Noatak watersheds (**d**). Winter precipitation included November through March.

envelopes of established treelines and colonists implies caution when predicting range expansions on the basis of gridded climate data[1,22–24].

The results here suggest that established treelines may not provide appropriate examples of patterns and processes involved in forest advance, corroborating studies in central Alaska where experimentally elevated temperatures increased growth more at locations above than at the treeline[45]. The metrics indicating more rapid growth in colonists than in individuals from established treelines also parallel measurements showing greater productivity at forest margins than within forests of Alaska[46,47].

## Regional extent

The AOI population is not the only one to recently colonize a tundra basin in Arctic Alaska. Within northwest Alaska watersheds distant from the AOI (Wulik basin and uppermost Noatak tributaries), we discovered four other spruce populations that during the last three decades have dispersed across mountains forming the boreal–Arctic divide (Fig. 1a). Advancing at a median speed of 4.9 km per decade from established treelines 4.8–25.5 km away, these nascent populations of a few (1–3 encountered per site), small (15–60 cm), young (17–32 years old) individuals may represent the first post-LGM colonists to arrive in their respective Arctic watersheds[28].

Spanning 20° of longitude (143–163° W) and equivalent to 20% of white spruce's entire east–west range (63–163 ºW)[26], small populations of 1 to >70 individuals grow vigorously, well beyond established treelines (mean of 4.4 km distant, $n = 34$ watersheds; Fig. 1a). The fewest colonies occurred in the eastern Brooks Range (mean of 2.4 km distant, $n = 4$ watersheds), where winter precipitation is lowest. Colonists most distant from established treelines occurred in the west (mean of 6.3 km distant, $n = 9$ watersheds), where winter precipitation is highest. A larger number of colonized watersheds were found in the central

range (mean of 4.0 km distant, $n = 21$ watersheds). In one instance, 150 km east of the AOI and 25 km from the triple divide of the Noatak, Kobuk and Koyukuk basins, we resurveyed a 4 km² valley previously censused above and beyond the established treeline[48]. There, spruce had increased in number by a factor of 12 over 43 years, doubling every 1.2 decades and increasing in height from a population of juveniles ≤1.2 m tall to one including adults 8 m tall and bearing cones.

At the northeastern range limit of white spruce in maritime Labrador[12] and near Hudson Bay[15] in Canada, field studies have also reported forest advance during the twentieth century. By contrast, remote sensing studies in the continental area between Hudson Bay and the McKenzie River Delta have found leading-edge disequilibrium[20] and even forest retreat[18] during the last half of the twentieth century. The most rapid modern expansion of a boreal tree into Arctic tundra so far reported is for mountain birch (*Betula pubescens*), a small deciduous–broadleaf tree advancing across Fennoscandia in northwest Eurasia at rates similar to those seen for white spruce here[5]. Arctic-wide greening trends from high-resolution satellite imagery during 1985–2019 also suggest a global boreal biome shift northwards as temperatures continue to rise[49]. Over time, warming—both directly[16] and indirectly[34–37]—will support higher-latitude tree establishment, growth, reproduction and dispersal, particularly with increased southerly winds, greater snowfall and increases in nutrient availability, all induced by warming.

## Conclusions

The proliferation of spruce in the twentieth and twenty-first centuries we describe represents a climate-driven invasion of Arctic tundra occurring at >4 km per decade, matching the post-LGM rate of white spruce migration out of glacial refugia[7]. Previous population reconstructions of spruce at their range limit found rapid infilling during the latter half of the twentieth century[9–15,18,20,21], but the incremental expansions documented[5] in most studies (<0.1 km per decade) are an order of magnitude lower than here. The orders-of-magnitude difference between forest migration and rates of isotherm movement leave vegetation increasingly out of equilibrium with climate[5]. How modern forests will respond in the face of changing climate remains poorly understood, and studies identifying instances of rapid advance may suggest mechanisms that increase LDD, facilitate sapling recruitment and increase sexual reproduction.

Decades behind the range expansion of tall shrubs[50], conifers may be on the verge of a stochastic, climate-driven invasion of tundra after centuries of stasis. Seedlings have established populations undetected by remote sensing in the relatively inaccessible Arctic, where beds of knee-high spruce will become trees of 4–5 m in height with heavy cone crops in 50 years. This increasing Arctic tree cover is accelerating as a consequence of and feedback to climate changes that will shift subsistence resources available to Arctic peoples[1], decrease habitat for migratory species[1], reduce land-surface albedo[2–4] and redistribute carbon stocks[4], all with global implications[1–4].

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

## Methods

White and black spruce are the dominant conifers at Arctic treelines and the boreal forest–tundra ecotone more generally in North America, with white spruce dominating on better drained sites. White spruce reaches its northwestern-most limit in Alaska, USA, at 68.1° N, 163.2° W. For comparison, the northeastern range extent of the species[26] is Labrador, Canada, at 57.9° N, 62.5° W (ref. [12]), giving an east–west range of >100° in longitude. Of the approximately 6,500-km-long northern boundary of white spruce in North America, 10–15% is located in Alaska's Brooks Range, where white spruce is the dominant treeline tree.

### Study area

The 1,000-km Brooks Range is a high-latitude mountain range dividing Arctic tundra from boreal forest in Alaska. The mountains and nearby lowlands are notable for their wilderness character, protected as a near-contiguous conservation area of >150,000 km². In the east between the Arctic Ocean's Beaufort Sea and the uppermost Yukon River basin, the range is cold and dry, reaching 2,736 m above sea level. The south slope of the eastern Brooks Range is included in Alaska's Northeast Interior climate division, where precipitation is among the lowest in the state[51]. Descending to the Chukchi Sea in the west, the range is included in Alaska's West Coast climate division, where precipitation is the highest in northern Alaska[51].

The Noatak and Kobuk rivers flow in their entirety above the Arctic Circle, draining the western Brooks Range. Both rivers empty into the Chukchi Sea near Kotzebue, Alaska (Fig. 1a). The Baird Mountains of the southwestern Brooks Range separate the Kobuk from the Noatak basin, and the De Long Mountains of the northwestern Brooks Range separate the Noatak from the river basins of the North Slope and from the Wulik basin, located northwest of the Noatak basin. The lower basins of the Noatak and Kobuk rivers are included in the West Coast climate division, with greater precipitation, warmer winters and cooler summers than in the Central Interior climate division and greater precipitation and warmer temperatures than in the North Slope climate division[51]. The upper basin of the 700-km Noatak River lies at the intersection of all three climate divisions, which warmed from 1949 to 2012; December–January precipitation increased from 1949 to 2012 in the West Coast climate division, as did North Slope winter precipitation from 1980 to 2012 (ref. [52]).

The Noatak River basin is entirely protected within federal conservation units. Its vegetation includes dwarf, low and tall shrub tundra communities that cover about 60% of the 33,000 km² basin[53]. Tussock sedge tundra covers another 30%, and wetlands and barrens cover most of the remainder. The main valley and tributaries along the lowest 200 km of the Noatak River support stands of white spruce, typically associated with a deeper active layer or an absence of permafrost. The treelines bounding these forests have long been identified as the northwest range extent of white spruce[26].

The upper Noatak basin, a 500-km reach, is underlain by extensive continuous permafrost[54]. It has been considered empty of spruce since US Geological Survey (USGS) geologist Philip Smith explored the Kobuk, Alatna and Noatak rivers by canoe in 1911 (ref. [55]). The adjacent Kobuk and Alatna river basins support boreal forests of black and white spruce, paper birch and aspen along much of their lengths. By surveying transects at and beyond hydrological divides separating the Noatak, Wulik, Kobuk and Alatna river basins, as well as further east in the Brooks Range (Fig. 1a), and informed by very high-resolution satellite scenes (Fig. 1b and Supplementary Figs. 1–13), we documented the locations of over 7,000 individual spruce colonists (Extended Data Fig. 1b–d and Supplementary Figs. 1–3). Overall, we traversed 22° of longitude (141–163° W) in the field, mostly along the treeline from Canada to the Chukchi Sea, locating dozens of populations of colonizing spruce (Fig. 1a) above alpine and beyond Arctic treelines (see 'Regional extent of colonization').

The primary AOI (Fig. 1a) included the USGS Hydrological Unit Code (HUC) 10 watersheds Kaluich, Cutler, Amakomanak and Imelyak located in the HUC 8 Upper Noatak Subbasin. However, we also documented (longitude, latitude, distance from established treeline) fast-growing, healthy spruce well beyond established treelines within six additional western Arctic watersheds, each separated by over 30 km in the western Brooks Range and 80–200 km distant from the AOI. These populations are within the far upper reaches of the Noatak basin (Lucky Six Creek, 67.594° N, 154.858° W; Kugrak River, 67.428° N, 155.723° W; Ipnelivuk River, 67.552° N, 156.293° W; upper Wrench Creek, 68.251° N, 162.617° W); 25 km northwest of the nearest established treeline and outside the Noatak basin in the Wulik River valley (68.120° N, 163.219° W); and along the Chukchi Sea coast (67.041° N, 163.114° W). We also note that, in the central Brooks Range, humans have actively or inadvertently disseminated spruce seeds and juveniles on the North Slope, with individual white spruce germinating and surviving there for at least 20 years[37,56].

### Patterns of expansion

**Digitizing spruce shadows.** We used cloud-free Maxar Digital Globe WorldView-1 and WorldView-2 satellite scenes (WV; https://evwhs.digitalglobe.com/myDigitalGlobe/login) of snow-covered landscapes from three missions in early spring 2018, a near-record year for snow depth in northwest Alaska (Fig. 1b, Extended Data Table 1 and Supplementary Figs. 1–13). Ground sample distances of 0.47–0.5 m, a root-mean-squared error of 3.91–3.94 m and off-nadir angles of 5–25° with low sun-elevation angles of 18–27° provided clear images from which to digitize the lengths of individual spruce shadows and identify their locations (Supplementary Information sections 1.2 and 1.3). One technician (S. Taylor), supervised in quality assurance and quality control (QAQC) by R.J.D., digitized 5,986 shadows (densities in Extended Data Fig. 1b, locations in Supplementary Fig. 1) on GEP using WV images as super-overlays. The technician identified all spruce shadows across the imported image tiles and then digitized them as line segments from base to shadow tip.

The super-overlays degraded the imagery somewhat, making small tree shadows more difficult to distinguish from snowdrift, rock or shrub shadows (Supplementary Figs. 5 and 6). We suspect that many trees in the height class of 2–3 m were missed. These line segments, saved as .kml files, were imported into R (v.4.1.1)[57] using the sf package[58], where the length of each line segment was calculated and the coordinates of the shadow's base were identified. The line segment lengths were used to estimate tree heights, and the coordinates were used in nearest-neighbour calculations and extractions of gridded data values. We estimated snow depth at 2.5–3 m because geolocated trees measured as ≤2.5 m in the field (see below) did not appear on imagery. We observed some trees taller than 2.5 m with no visible shadows on imagery, possibly buried in deeper snow or growing in shadows cast by terrain at the time of image capture. Thus, our estimates of adult populations may be underestimates, although there were also errors of commission where shrub shadows were mistakenly classified as spruce (see following).

**Digitizing and field validation.** To estimate identification accuracy (Supplementary Information sections 1.2 and 1.3) among the 1,971 digitized shadows used for population reconstruction (enclosed by red rectangles in Supplementary Figs. 1–4), we visited 157 shadow locations first identified on imagery (8% of the 1,971) and located in the field with the built-in GNSS of late-model Apple iPhones (models 12 Pro Max, 12 Pro and second-generation SE) with positional accuracy in the open landscapes estimated at 3 m. At these 157 locations, 11 shadows were cast by very tall willows (7%). Of the 146 shadows confirmed as trees, 2 were dead (1%) and 1 had a recently broken top with green foliage on the ground. We added the length of the broken top to the standing height measured with a laser range-finder. Trees that were collinear in

the solar azimuth at image capture contributed to errors of omission. The tree standing to solar azimuth obscured others as overlapping shadows fell in line, generating both errors of omission and an overestimate of the height of the first tree in the series. Six trees shadowed in three instances by what we identified on imagery as single shadows fell in this category. An additional three trees were missed during digitizing, also going unnoticed during QAQC, and were discovered in the field when matching shadows with trees. Supplementary Information section 1.3 provides details and a confusion matrix.

In summary, 157 trees were expected from digitized shadows and 155 were found in the field. Applying the accuracy of the count overall suggests that 1,945 trees would better estimate the reconstructed population. Across the AOI, the total adult count of 5,988 shadows may represent 5,910 trees. Moreover, in so far as our estimates of ages based on tree heights are predictive, perhaps 2% of the 'trees' in our reconstruction are not a single tree casting a long shadow, but 2–3 younger, collinear trees. Thus, our estimate of past populations may be slightly biased to older trees, implying that the population growth rate may be slightly higher than estimated. However, the slightly fewer trees than shadows would suggest that the growth rate is lower. The relative size of these errors appears minor, and we did not incorporate them into the analysis, which seems to us robust and perhaps conservative in adult abundance estimates owing to image degradation with GEP super-overlays and other errors of omission. This study would have benefited from less image degradation using dedicated geographic information system (GIS) or image software. However, the low cost, simplicity and convenience of GEP was appealing for the large-scale digitizing.

Returning from the field with individual tree data, R.J.D. displayed digitized shadow points together with field points on GEP, visually matching each field point to the nearest shadow, conditional on relative congruence between shadow size and tree height. This required care in clumps of trees with varying heights (example in Supplementary Information sections 1.2–1.3). The relative patterning of field points compared with shadows and the lengths of shadows compared with tree heights in these cases provided some measure of confidence in attribution.

We made field expeditions to six study areas within the extent of the WV imagery we used for digitizing, three within the 'simulated population area' rectangle in Extended Data Fig. 1a (red rectangle in Supplementary Figs. 1–4) and three study areas further east (Extended Data Fig. 1c and Supplementary Fig. 2). Among-area variability was apparent in snow depth, terrain slope relative to the solar azimuth at the time of image capture and the solar-elevation angle itself because of the timing of image capture. The variability was identified, calculated and applied on the basis of geographic variability in the heights of trees casting shadows and from the slope and intercept of a mixed-model linear regression of field-measured height on digitized shadow length (see below).

**Field surveys.** We validated species and heights of spruce casting shadows within the AOI along 403 km of ground transects. Our sampling did not appear spatially biased when compared with imagery as measured by proximity to a remote fixed-wing-aircraft landing site. Four field campaigns focused on three objectives in watersheds that were within or adjacent to the Noatak basin but did not have established treelines visible on WV growing season scenes: (1) to locate and document colonists at the geographic range boundary of white spruce; (2) to verify the locations of a sample of trees suggested by imagery in the AOI; and (3) to collect ecological measurements germane to white spruce range expansion. For adults (trees ≥2.5 m), datasets included height above ground ($n = 340$), diameter at breast height (DBH (-1.4 m); $n = 296$), CAG ($n = 17$), foliar nutrient content ($n = 17$), basal increment cores taken ≤20 cm above the ground ($n = 140$), tall shrub abundance within 5 m of sampled adults ($n = 246$), counts of juveniles within 5 m of sampled adults ($n = 250$), abundance class of cones ($n = 339$) and status

of adults (live, $n = 340$; dead, $n = 8$). Of the dead adults, seven of eight were standing and largely without bark, with a median height of 4.1 m. The fallen dead tree was 6.2 m long with a DBH of 13.4 cm; all bark and limbs to fine branches remained. Only one dead adult, 4.1 m tall with a DBH of 4 cm, showed signs of decomposition with shelf fungus on the stem and decomposed limbs on the ground. Five juveniles ≥1.5 m tall had been stripped of their bark and all but their uppermost branches by apparently either porcupine (*Erethizon dorsatum*) or snowshoe hare (*Lepus americanus*). Anecdotally, we recorded other signs and possible causes of damage such as wind, bear (*Ursus arctos*), caribou (*Rangifer tarandus*) or struggling growth such as layering, stunted krummholz or clonal reproduction, although these growth forms were nearly totally absent.

Field measurements for $n = 770$ juveniles located in the AOI and presented here included overall height, height above ground of bud scars representing 2015–2020 height ($n = 302$), damage and status. We used these measures to estimate age to increment core of adults (Supplementary Information section 2) and the RGR of juveniles (Supplementary Information section 3).

**Range expansion analyses.** Digitized established treelines (DETs) used here were downloaded as CTM_Treeline.kml from https://arcticdata.io/catalog/view/doi:10.18739/A2280506H. Ref. [34] describes drawing DETs on very high-resolution satellite imagery such as WV and Quick Bird. We clipped DETs to the four USGS HUC 10 watersheds within the HUC 8 Middle Kobuk subbasin and adjacent to the AOI (see 'Environmental conditions' below). The coordinates of the vertices for the clipped DETs provided the 3,366 locations of established treelines.

We used the rdist.earth() function in the R package fields[59] to identify the nearest neighbouring mapped adult and juvenile colonists in the AOI and DET vertices in adjacent Kobuk watersheds (Supplementary Information sections 1.8 and 1.9). Using the coordinates of nearest neighbours, we calculated differences in latitude as latitudinal displacement. Displacement north equalled the product of latitudinal displacement and 111.32 km, the distance between 67° and 68° N along 157.6891° W, which splits the AOI. Displacement in elevation was found by extracting from Interferometric Synthetic Aperture Radar (IFSAR) Alaska 5-m digital elevation models (DEMs) the elevation of DET vertices, mapped adults and mapped juveniles using the extract() function in the raster R package[60] and then subtracting the elevation of the nearest neighbours from focal adults and juveniles. When geolocated adults or juveniles had estimated establishment years (see 'Individual growth' below), we calculated movement rates as the difference between the establishment year of an aged tree and the establishment year of the oldest tree sampled (1901, year of founding) as the denominator and displacement (difference in metres above sea level, kilometres or degrees of latitude) as the numerator (Supplementary Information sections 1.19–1.21). To time the progression of spruce away from DETs, we also binned establishment year by decade as decadal class, identifying within each decadal class the maximum displacement in kilometres north of and elevation in metres above (or below) nearest neighbours.

## Population growth

From the 5,986 spruce shadow lengths within the AOI (Extended Data Fig. 1b and Supplementary Fig. 1) that we digitized from snow-covered scenes of DigitalGlobe WV imagery (Extended Data Table 1), we identified a sample of shadows stratified by length and cast by spruce that we located with GNSS-equipped late-model iPhones. We measured the height of $n = 260$ trees using a laser range-finder (LTI TruPulse 200) and/or a smartphone app (Arboreal Tree on iPhone 12 Pro and Pro Max with laser scanners) and collected $n = 122$ basal cores from individuals ≥2.5 m in height, then matched to shadows on imagery as described above (see 'Digitizing and field validation'). Using the relationship between height and shadow length and the probability distribution of establishment year for the 122 cored trees identified within five height classes

(Extended Data Fig. 2b), we simulated population growth within two contiguous sub-watersheds (the 135 $km^2$ 'simulated population area' in Extended Data Fig. 1a; western portion in Extended Data Fig. 2a; red rectangles in Supplementary Figs. 1–4; details in Supplementary Information section 4). These sub-watersheds contained $n = 1,971$ shadows cast on 26 March 2018. We treated these shadows as single spruce but recognize that they include as many as 138 willows (7%) and calculate an additional 118 (6%) spruce missed either by digitizing omission or by collinearity (Supplementary Information sections 1.2 and 1.3). Incorporating these errors together would not change the outcome of the simulations enough to change the doubling time of the population by more than a few percent.

**Estimates of tree height from shadow length.** On a flat landscape covered uniformly in snow, the total height $H$ of a tree equals snow depth $S$ added to the product of shadow length $L$ on the snow surface and the tangent of solar-elevation angle $\alpha$, as $H = S + L\tan(\alpha)$. However, because both the relative solar elevation and snow depth vary with terrain, we used a linear mixed-effects model (lmer() in the lme4 R package[61]) of height on shadow length (random factor of sample area with six levels), interpreting the fixed-effects intercept as the average snow depth (mean ± s.e. = 2.84 ± 0.14 m, $t = 20.29$) and the regression coefficient as the average tangent of solar elevation relative to the terrain slope (0.27 ± 0.04 m m$^{-1}$, $t = 6.96$; details in Supplementary Information sections 4.1 and 4.2).

Using these fixed-effects estimates and the random-effects covariance matrix, we applied Monte Carlo sampling to estimate the 1,971 heights with each run of the simulation, thereby propagating the error in height estimates. These 1,971 heights were then binned into five height classes with 0.5-m intervals from 4–5.5 m and with ≥1-m intervals from 3–4 m and 5.5–7 m (details in Supplementary Information sections 4.3 and 4.4). Height classes deduced from the shadow measurements were in some cases only 0.5 m in width. Because the mean snow depth (the intercept in the mixed-effects model) differed by more than this from one part of the study area to another (BobWoods, GaiaHill and BuffaloDrifts in Supplementary Information sections 4.1 and 4.2), this approach may have introduced systematic misclassification between locations. While applying a Monte Carlo model with coefficients drawn randomly using the mvrnorm() function from the MASS package in R with the random-effects covariance matrix was meant to alleviate this, we also ran the simulation with three uniform height classes with a wider interval (1.3-m width, for classes of 3–4.3 m, 4.3–5.6 m and 5.6–7 m).

**Estimating population-scale establishment year.** We estimated establishment years for each of the 1,971 trees (Supplementary Information sections 4.3 and 4.4). We did so by using the establishment year distributions by height class as Gaussian kernel densities for the 122 aged adults binned into the five height classes defined above (Extended Data Fig. 2b). Kernel density estimates were constructed using the function density() in R with options bw = "SJ" as the smoothing bandwidth, $n = 107$ as the number of consecutive establishment years, from = 1897 as the earliest year and to = 2004 as the latest year. For each of the 1,971 estimated heights binned into height classes, an establishment year was drawn (with replacement) from the corresponding kernel density distribution. We interpreted the total number of individuals in each establishment year as 'recruitment by year' into the population of survivors that we had digitized on the 2018 imagery. Sorting and cumulatively summing recruitment by year gave what we interpreted as population size ($N$) for each year ($t$) for trees that survived to 2018. Resampling in this manner for 1,000 runs, each time fitting exponential growth equation $N(t) = N_0 e^{k(t-1900)}$ using nls() in R and then averaging the population RGR, provided population doubling time as ln(2) divided by mean $k$. The simulation was run again using three height classes, each of 1.3 m in width. The resulting mean doubling time was unchanged, but variability increased (Supplementary Information section 4.6).

## Individual growth

**Current annual growth and foliar chemistry.** In autumn 2019, we collected current-year lateral branch tips on the west and east sides of each sampled spruce ($n_1 = 17$ adult colonists and $n_2 = 457$ adults at established treelines) at 1.4 m above the ground. Current annual branch growth was measured on 2–6 branches per spruce from the previous year's bud scar to the tip of the branch. The number of samples varied, ensuring sufficient mass for foliar chemical analysis. Established treelines were sampled for adult foliage in 12 watersheds of the Noatak, Kobuk and Koyukuk river basins where we have ongoing experiments. At these sites, we used a replicated nested plot-based design (Extended Data Table 3). Colonist foliage sample locations ($n = 8$) in the upper Noatak basin were widespread across three watersheds. At each location, except the upper Noatak where 1–3 spruce per location were sampled, we sampled $n = 5$ white spruce separated by ≥10 m at a DBH of 8–12 cm. Needles from each branch tip were pooled by individual, dried for 48 h at 60 °C and weighed. Needles of individuals were pooled by treeline location after grinding to powder using a steel ball mill grinder (Mini-Beadbeater, Biospec Products) and subsampled for chemical analysis. Foliar N and $^{15}$N isotope were analysed for one subsample run on an Elemental Combustion Analyzer (Costech, 4010) coupled to an isotope ratio mass spectrometer (Delta Plus XP, Thermo Fisher Scientific) at the University of Alaska Anchorage Environment and Natural Resources Institute Stable Isotope Laboratory. Foliar P was measured for another subsample by the Pennsylvania State College Analytical Services Lab using the acid digestion method and analysed by inductively coupled plasma emission spectroscopy[62].

**Juvenile RGR.** Several results presented here depend on juvenile vertical height growth during 2015–2020, which we assumed followed $h(t) = h_{2015} e^{(RGR\ t)}$, where $h(t)$ is height above ground for year $t$ after 2015, $h_{2015}$ is the height above ground in 2015 and RGR is the relative growth rate (Supplementary Information section 3). We used juvenile RGR in three contexts: (1) as a means of estimating establishment year in juveniles (Supplementary Information section 3.3); (2) as a metric of growth for comparison between colonist and established treeline juveniles (Supplementary Information section 6); and (3) to estimate the establishment year of cored trees (see second paragraph in 'Dendrochronology' below and Supplementary Information section 2).

To estimate the RGR for each of 505 juveniles ($n_1 = 300$ juveniles from $m_1 = 4$ colonist populations and $n_2 = 205$ juveniles from $m_2 = 14$ established treelines; Extended Data Table 2), we measured the heights above ground ($h$) of the six uppermost bud scars in 2020, representing height increments in 2016–2020, the five consecutive years with the warmest mean daily July air temperature on record for Kotzebue. RGR in each juvenile was calculated as the regression slope of $\ln(h(t))$ against $t$ (mean $R^2 = 0.99$ for 300 colonist regressions and 0.98 for 271 established treeline regressions; Supplementary Information section 3.4).

To estimate the establishment year of juveniles, we used RGR to back-calculate $T$, the years required for an individual colonist to grow from 2 cm to $h_{2015}$, as $T = \ln(h_{2015}/2)/RGR$. By subtracting $T$ from 2020, we estimated the establishment year of each juvenile (Supplementary Information section 3.3).

RGR values for colonist and established treeline juveniles (Extended Data Table 2) were compared using a linear mixed-effects model with field site ($m = 24$) as a random intercept, ln(RGR) as the dependent variable, $\ln(h_{2015})$ as a covariate to capture allometric growth and population (colonist or established treeline) as the fixed factor of interest (Supplementary Information section 6). Using the lmer() function of the lme4 package[61] in R with REML = F, we found that the Akaike information criterion (AIC) for the interaction model was lower than that for the corresponding additive one (ΔAIC = 22, likelihood ratio test $\chi^2 = 24$, degrees of freedom = 1, $P < 0.0001$). Applying lmer() with REML = T,

we report the *t* score of the interaction between population and ln($h_{2015}$) as a test of the null hypothesis that mean RGR in colonist and established treeline juveniles was equal (Supplementary Information section 6.2).

**Dendrochronology.** We collected one basal increment core per tree for *n* = 140 trees at core height *C*, where 2 cm ≤ *C* ≤ 20 cm above ground (median, 5 cm). We selected trees with generally symmetric crowns in open locations with bark encircling the entirety of their mostly circular bole at core height. Increment cores were used to age individuals and to compare radial stem growth to mean daily July air temperature from Kotzebue during the period from 1990 to 2020. Cores were mounted, sanded and scanned at 1,200 d.p.i. We imported scans into CooRecorder (https://www.cybis.se/forfun/dendro/index.htm) for ring width measurements. We visually cross-dated cores[63] and then checked cross-dating using COFECHA[64] and CDendro (https://www.cybis.se/forfun/dendro/index.htm). In cores lacking pith, we used CooRecorder's pith locator tool, which estimates distance and years to pith using the width and curvature of the innermost tree ring widths. Tree establishment year (*Y*) was estimated by subtracting the establishment age at core height from either the pith year or the year of the innermost tree ring minus the estimated years to pith. Growth age to core height was estimated using the height and age relationship from *n* = 83 colonists with 4 cm ≤ $h_{2015}$ ≤ 20 cm and RGR calculated as described (see 'Juvenile RGR' above). For each of these 83 juveniles, we calculated *T* = ln($h_{2015}$/0.9)/RGR as years to 0.9 cm. The value *T* formed the dependent variable in a linear mixed model (*m* = 4 field sites as random factor) with ln($h_{2015}$) as the predictor variable. The fixed effects gave *T* = −7.31 + 10.46ln*C*, where *C* is core height (Supplementary Information section 2).

Raw tree-ring measurements were processed (Supplementary Information sections 7.3–7.5) to yield time series as BAI and as residuals from AR models[65,66]. BAI compensates for a possible decrease in ring growth increment with tree age by combining the size of the tree (radius, *R*) and its growth increment (Δ*r*) as BAI = 2π*R*Δ*r*. Although BAI is appropriate for direct analysis of growth, analyses of climate–growth relationships should also account for autocorrelation found in tree-ring series[65,67]. AR ('pre-whitening') results in series that approximate white noise with means of 1. The AR order of best fit was determined for each series by AIC score (AR order of 1 was most common). This method removes all but the high-frequency (interannual) variation in the series, which can then be compared to climate series. We also applied the AR approach to the July mean temperature data, but two common methods of selecting AR models (the Yule–Walker and maximum-likelihood methods) indicated that these data already approximated white noise (AR order of 0; Supplementary Information section 7.2). Thus, we retained the raw temperature data in subsequent analyses. All tree-ring detrending and standardizations were performed using the R package dplR[68].

We assessed relationships between ring indices and July temperature using Pearson's product-moment correlation of both ln(BAI) and AR with Kotzebue mean daily July air temperature from 1989 to 2019 using the cor.test() function in R with a two-tailed significance test (Supplementary Information section 7.5). We grouped the 140 increment cores as 'juveniles' <30 years old (*n* = 15) or 'adults' ≥30 years old (*n* = 125) for correlation with Kotzebue July temperature from 1989 to 2019. Adults must allocate structural carbohydrates to both growth and reproduction, whereas juveniles do not allocate to reproduction, suggesting that the strength and direction of radial growth responses to temperature may differ between age classes.

### Environmental conditions

We used USGS HUC 10 and HUC 8 hydrological basins and watersheds to delineate our study AOI (https://water.usgs.gov/wsc/a_api/wbd/subbasin19/19050401.html; accessed from https://apps.nationalmap.gov/downloader/#/). The AOI was located in HUC 10 watersheds

Kaluich, Cutler, Amakomanak and Imelyak of the HUC 8 Upper Noatak Subbasin. The DETs nearest the AOI were located in the Redstone, Miluet, Akillik and Hunt watersheds (HUC 10) of the Middle Kobuk Subbasin (HUC 8).

July temperature (Extended Data Figs. 3 and 6c) and November–March precipitation (Extended Data Fig. 6a) records from Kotzebue Airport (Fig. 1a) were accessed from the National Oceanic and Atmospheric Administration (https://www.ncdc.noaa.gov/cdo-web/search). July air temperatures (Extended Data Fig. 3) and October–April wind directions for the Kaluich (13 km west of the AOI; 758 m above sea level) and Imelyak (44 km east; 1,089 m above sea level) RAWS stations were accessed from the Desert Research Institute (https://raws.dri.edu/akF.html). Thirty-year (1980–2010) gridded (0.00833° resolution) mean July air temperature and November–March precipitation were accessed from the PRISM[69] climate group (https://prism.oregonstate.edu/projects/alaska.php). We summed the PRISM November–March precipitation data (the water equivalent of snow for those months) by pixel. All correlations using these data sources applied Pearson's product-moment correlation (*r*) with the cor.test() function in R and a two-tailed significance test. Strong, stable environmental lapse rates in July occur from Kotzebue to 1,200 m above sea level in the western Brooks Range (Extended Data Fig. 3 and Supplementary Information 7.2), making the instrumental temperature record there relevant to the AOI.

We obtained sea-ice cover in the Chukchi Sea from the National Snow and Ice Data Center (https://nsidc.org/arcticseaicenews/sea-ice-tools/). Time series of Chukchi Sea open water were derived from the file labelled 'Monthly sea ice extent, by region' (N_Sea_Ice_Index_Regional_Monthly_Data_G02135_v3.0.xlsx). We identified the overall maximum ice extent across years and months for the Chukchi Sea (maximum of 8.232 × 10^5 km² in April 2006) and then subtracted October ice cover values from the maximum and defined the difference as 'October Chukchi Sea open water' (Supplementary Information section 8.3). We chose October as snow from that month is the first of the season that generally remains throughout the winter.

Bivariate climate envelopes in the AOI were based on the PRISM[69] 30-year mean July temperature and summed 30-year mean November–March precipitation extracted within the four HUC 10 watersheds in the HUC 8 Middle Kobuk Subbasin and the four HUC 10 watersheds of the Noatak AOI (Supplementary Information section 8.1). To construct a bivariate climate envelope for DETs, we extracted the Middle Kobuk Subbasin gridded PRISM data using the coordinates of the DET vertices. For the colonist population, we extracted the gridded PRISM data within the AOI using the coordinates of all mapped adult and juvenile colonists.

DEMs for estimating the elevation of geolocated adults, juveniles and DETs consisted of 5-m-resolution digital surface models collected using IFSAR Alaska and downloaded using USGS EarthExplorer (https://earthexplorer.usgs.gov/).

### Regional extent of colonization

Range expansion requires a sequence of successful life history stages: seed dispersal, seedling establishment, sapling recruitment and adult sexual reproduction. Over 22° of longitude (141–163° W) of the Brooks Range, R.J.D. led field crews in search of ongoing range expansion by juveniles <1 m tall representing successful dispersal ('seedlings'), juveniles >1 m tall representing successful establishment ('saplings') and individuals >2.5 m tall representing potential reproduction ('adults'). The unit of sample was the watershed. Where one or more white spruce individuals ('colonist populations') were encountered >1 km beyond the established treeline, we recorded the location, age classes and presence of cones when possible. In watersheds of the uppermost Noatak basin and the Wulik basin, we also recorded both the total height of juveniles and the height above ground of the sixth bud scar from the tip to estimate RGR and so estimate age. We encountered three watersheds with tree island krummholz >1 km beyond the treeline but do not include these as colonist populations because clonal growth can be very old[9–19].

Of the 34 watersheds in which we encountered colonist populations >1 km beyond established treelines, 4 watersheds were located between 141° and 149.7° W (eastern Brooks Range), 21 watersheds were located between 149.7° and 156.3° W (central Brooks Range) and 9 watersheds were located between 156.3° and 163.3° W (western Brooks Range). Watersheds west of 150.5° W with colonists are shown in Fig. 1a.

In 2021, R.J.D. led a field expedition to a small watershed in the Koyukuk basin (Arrigetch Creek, 67.439° N, 154.090° W). The watershed had been purposely surveyed for juvenile white spruce above and beyond the treeline during 1978–1980 when seven juveniles 11–112 cm tall (six seedlings <1 m, one sapling ≥1 m) were located and mapped[48]. Our resurvey of upper Arrigetch Creek found 70 juveniles (52 seedlings, 18 saplings) and 19 adults. Near the mapped location of the two tallest juveniles in ref. [48], R.J.D. found a cone-bearing adult, as well as an additional eight cone-bearing adults elsewhere in the watershed, for a total of nine trees with cones among 19 adults up to 8 m tall. Four decades earlier, the tallest tree reported there had been 1.12 m tall.

## Reporting summary

Further information on research design is available in the Nature Research Reporting Summary linked to this article.

## Data availability

Data used for the analysis presented here are archived at the National Science Foundation's Arctic Data Center at https://doi.org/10.18739/A2X63B650. Source data are provided with this paper.

## Code availability

Code used for the analysis presented here are archived at the National Science Foundation's Arctic Data Center at https://doi.org/10.18739/A2X63B650.

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

**Acknowledgements** L. Berner, A. Brownlee, D. Cooper, P. Burns, A. Dahl, P. Dial, J. Ditto, S. Donahue, F. Restrepo, J. Geck, R. Koleser, J. Kramer, T. Matsuoka, F. McCarthy, B. Meiklejohn, S. Smeltz, D. Stephens, S. Taylor, B. Weissenbach, R. Wong, D. Wright and M. Zietlow assisted with remote sensing, GIS, field and/or laboratory work, and D. Nickisch, E. Sieh and J. Cummins provided safe transport. The US National Park Service permitted the research as NOAT-2021-SCI-0002, GAAR-2021-SCI-0004 and GAAR-2019-SCI-0002. We acknowledge awards NSF OPP-1748849 to P.F.S., NSF OPP-2129120 to R.E.H. and NSF OPP-1748773, Explorers Club Discovery Expedition Grant, Alaska NASA EPSCoR (80NSSC19M0062), and NASA through the Alaska Space Grant Program to R.J.D. and his students.

**Author contributions** R.J.D., P.F.S. and R.E.H. designed the study. R.J.D., C.T.M., P.F.S. and R.E.H. performed field work. P.F.S. and C.T.M. performed and/or supervised laboratory and dendroecology analysis. R.J.D. supervised and performed image interpretation. R.J.D. and C.T.M. prepared maps and figures. R.J.D. led calculations, modelling and manuscript writing with contributions, edits and redrafting from all.

**Competing interests** The authors declare no competing interests.

**Additional information**
**Correspondence and requests for materials** should be addressed to Roman J. Dial, Colin T. Maher, Rebecca E. Hewitt or Patrick F. Sullivan.

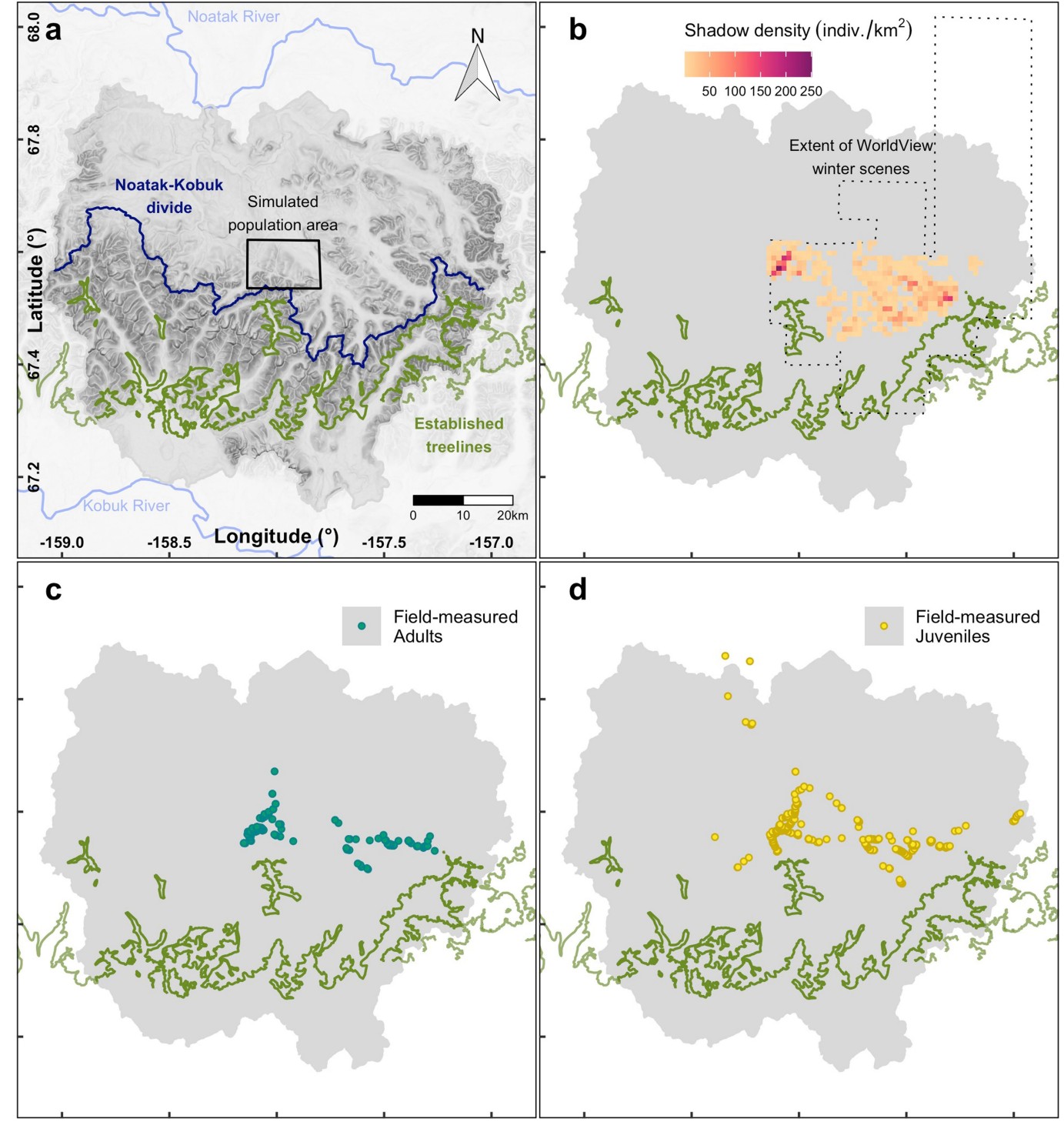

**Extended Data Fig. 1 | Adult and juvenile white spruce have colonized above and beyond established-treelines. a** Baird Mountains of the Brooks Range separate Arctic Noatak basin from boreal Kobuk basin. Noatak-Kobuk hydrologic divide shown as dark blue line. Established-treelines visible on World View (WV) summertime imagery shown as olive green line from reference 44. Noatak and Kobuk Rivers in light blue. Gray polygon boundary encloses four Noatak and four Kobuk watersheds of area of interest (AOI). Small, central rectangle encloses population simulated over time in Fig. 3a with adult spruce densities shown for western portion in Extended Data Fig. 2a.

**b** Density of spruce tree shadows as individuals per square-kilometer rasterized from shadows digitized on WV1 and WV2 snow covered scenes (Fig. 1b, SI Figs. 1, 4–13). Dotted outline shows extent of imagery used to digitize shadows of trees over approximately 2.5 m tall. **c** Adult white spruce (≥2.5 m or cone-bearing) measured for height and other ecological measures (including 140 increment cored for dendrochronology) geolocated in the field using GNSS. **d** Juvenile (<2.5 m) spruce geolocated in the field using GNSS. Two juveniles were found north of the Noatak River and near to but outside the AOI. They are included in the analysis.

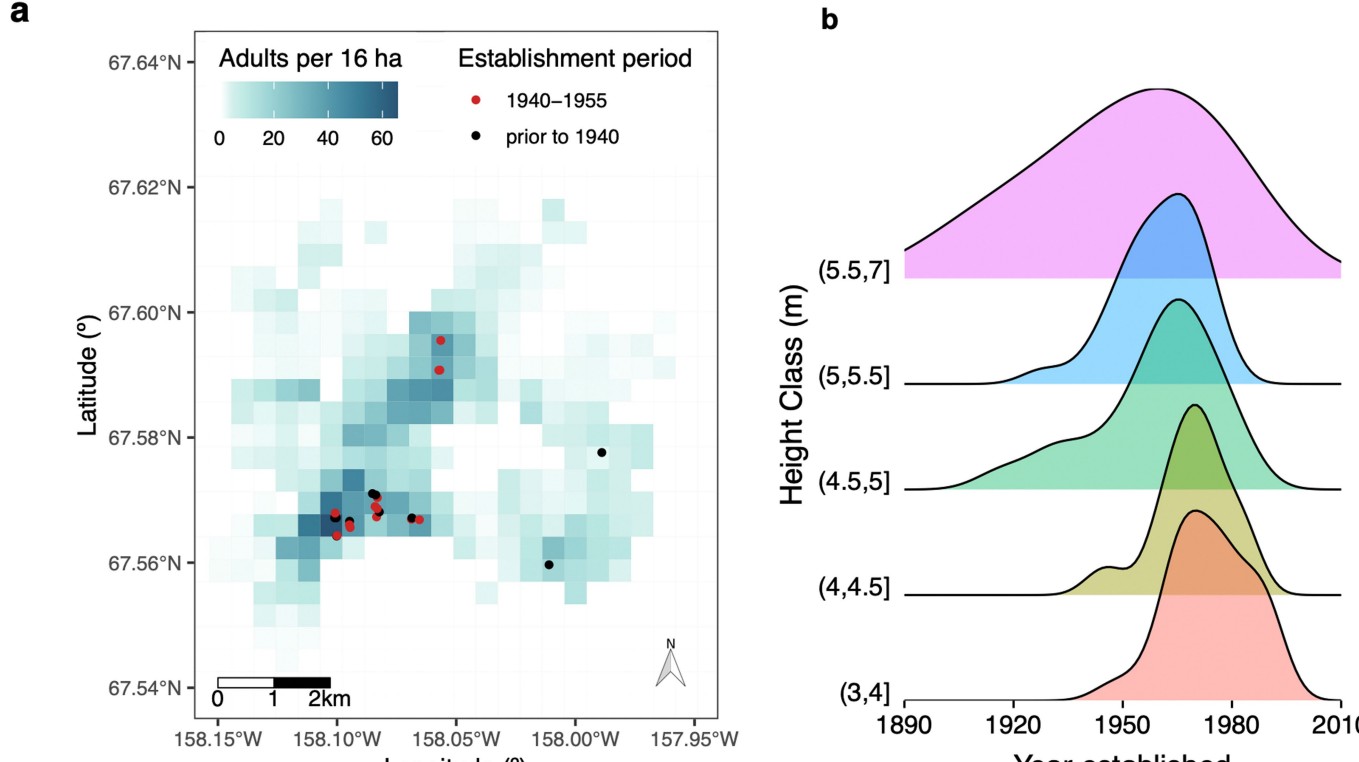

**Extended Data Fig. 2 | Density is higher near older colonists and ages vary by size-class. a** Stem densities (adults per 16 ha) calculated using a subset of the 1,971 digitized spruce shadow locations located within the western portion of the small, central black rectangle of Extended Data Fig. 1a and the western portions of the red rectangles in SI Figs. 1–4 and SI Fig. 11. Oldest aged trees from increment cores are superimposed as black (trees aged older than 1940 as establishment year) and red circles (aged to between 1940 and 1950). The pattern of ages and densities suggests recursive founding of small, local populations. **b** Year established by height class as Gaussian kernel densities using *n* = 122 increment cores, cross-dated and corrected for height at core. These distribution are used in reconstructing population growth in Fig. 3a (simulation in SI Calc. 4) of the 1,971 individuals in the region shown as the small, central black rectangle of Extended Data Fig. 1a. A sensitivity analysis was performed using three uniform height classes of width 1.3 m ((3-4.3m], (4.3-5.6m], (5.6-7m]) (SI Calc. 4.6).

# Dotted line : Mean July lapse rate = −4.99°C km$^{-1}$

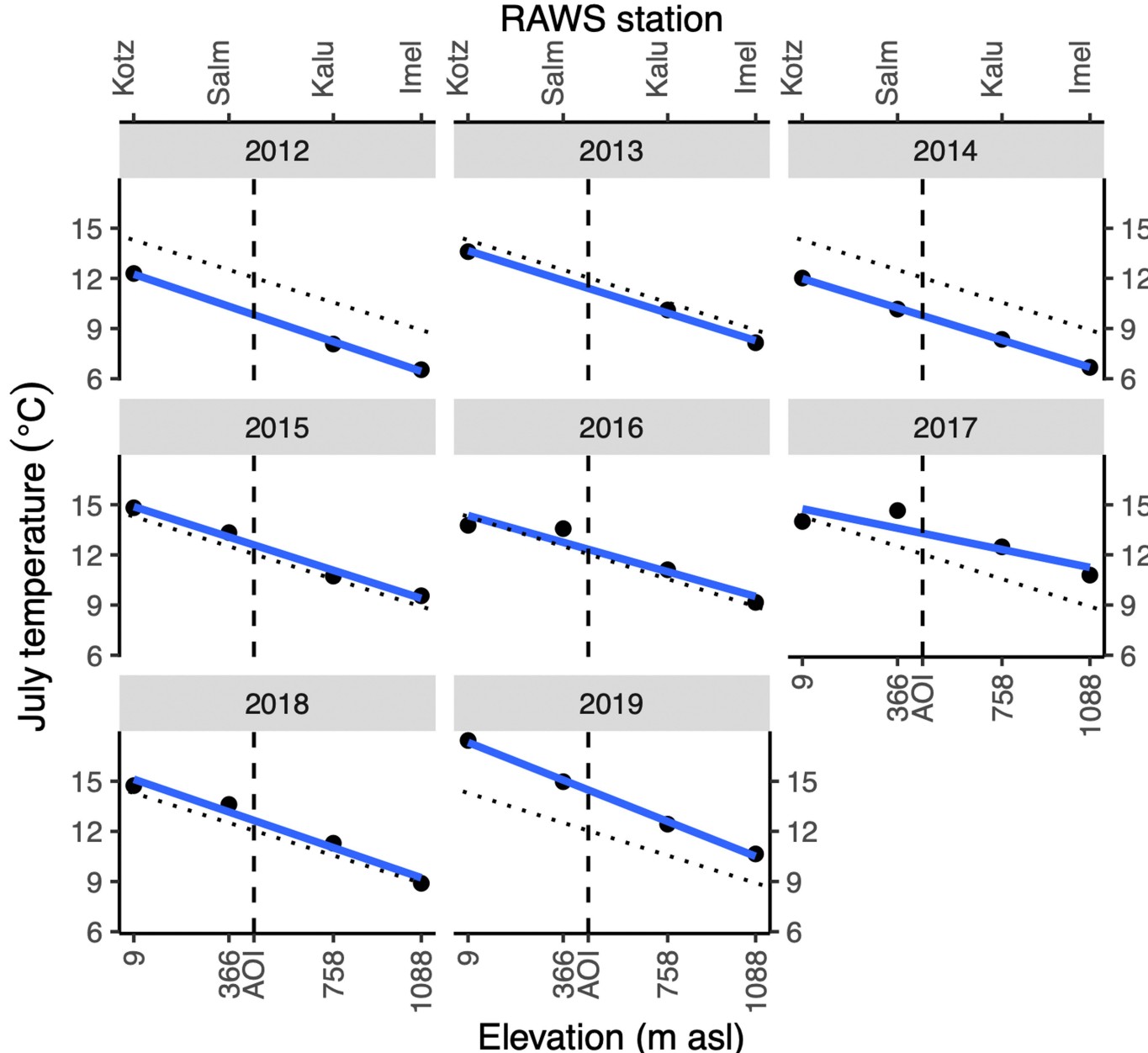

**Extended Data Fig. 3 | Kotzebue and Brooks Range Remote Automated Weather Stations (RAWS) show stable lapse rate.** Blue lines show regression of mean daily July air temperature on elevation by year (2012-2019) in northwest Alaska. Dotted lines give fixed effects slope and intercept of linear mixed effects model (year as random factor) of temperature on elevation at four weather stations: "Kotz" is Kotzebue airport and three remote automated weather stations (RAWS) within 250 km of Kotz are as follows: "Salm" is Salmon, "Kalu" is Kaluich, and "Imel" is Imelyak. The vertical dashed line at AOI is located at 460 m asl, the elevation of the oldest aged colonist in the AOI. Numbers on bottom axis give elevations of weather stations named on top axis. Mean (± se) July lapse rate is −4.99 (± 0.26) °C km$^{-1}$ (SI Calc. 7.2). Correlations of temperature from Kotzebue with temperatures from Salmon, Kaluich, and Imelyak RAWS were 0.967, 0.962, and 0.973 respectively (SI Calc. 7.1).

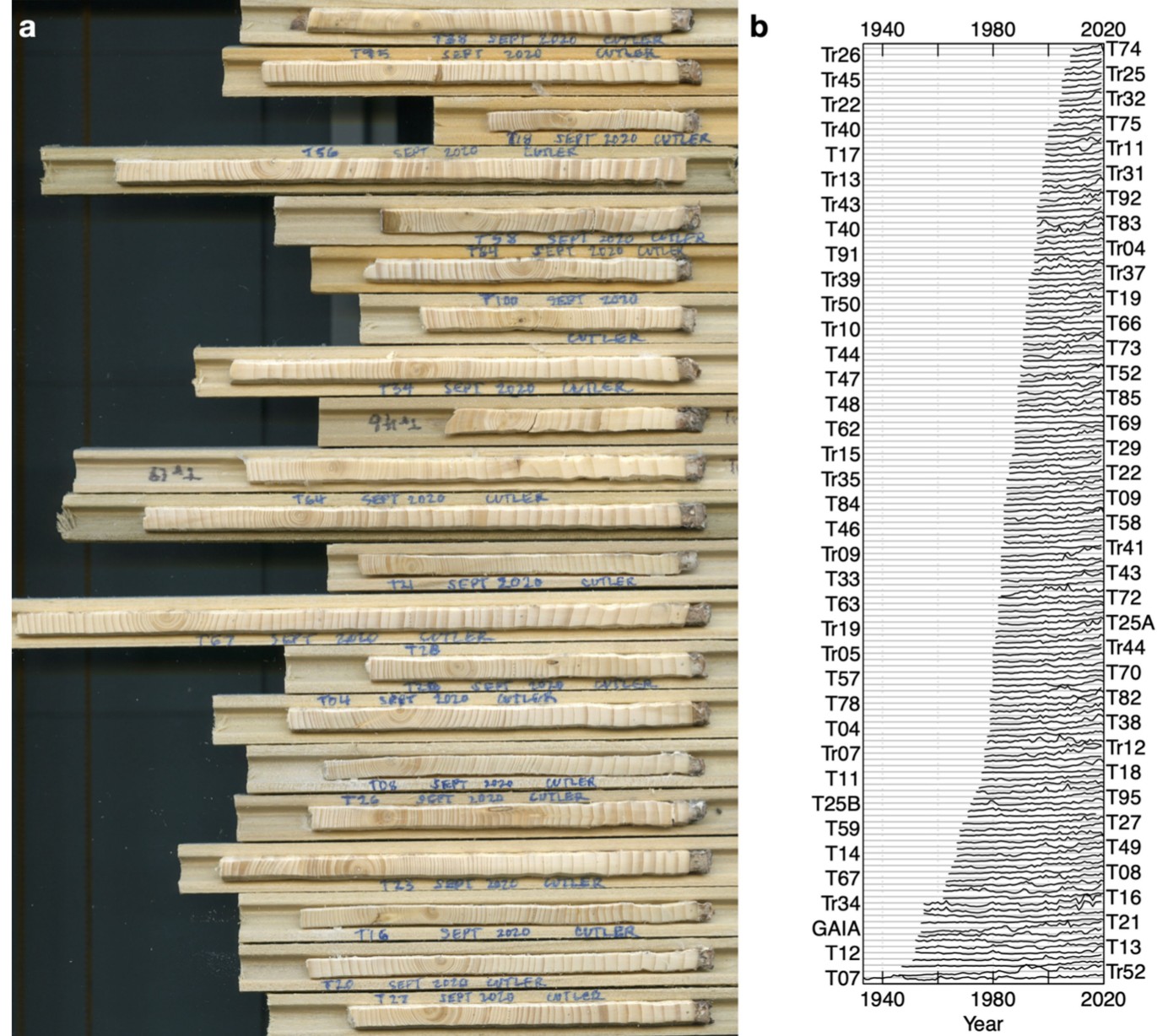

**Extended Data Fig. 4 | Raw ring widths increase across all ages during the warmest three decades of July temperatures recorded in Kotzebue, AK.** **a** Twenty-one of 140 total tree cores of colonists of all ages selected for inclusion of pith. Of all 140 cores 71% included the pith or were within an estimated five years of the pith. Most cores show accelerating growth over the most recent 30 years. **b** All 140 cores ranked by number of rings. Raw wing width on vertical and Gregorian year on horizontal axis.

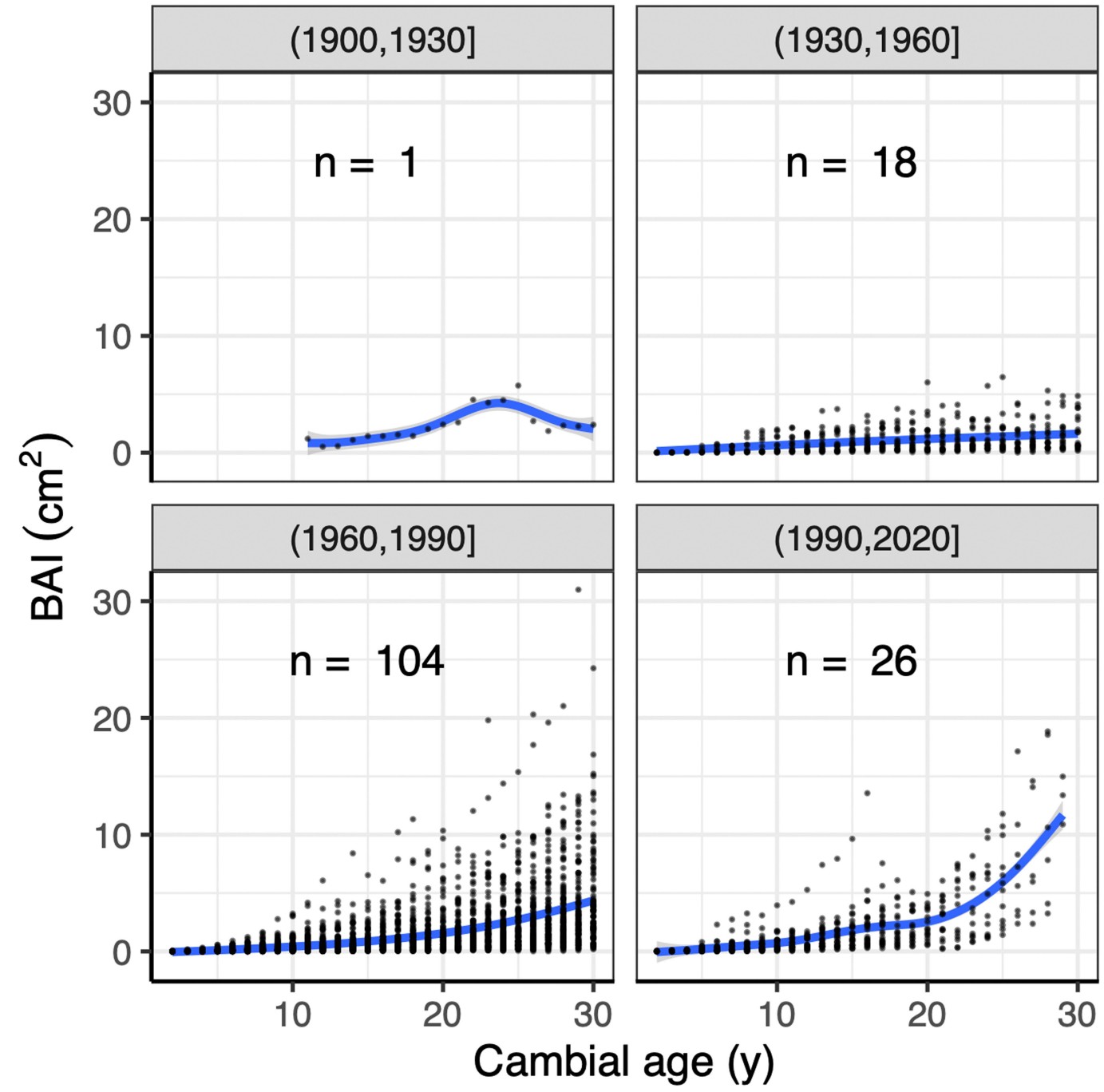

**Extended Data Fig. 5 | Growth as basal area increment (BAI) in juvenile cohort of the warmest three decades is more rapid than in juveniles of older cohorts.** Across 30-year cohorts given as (1900,1930], (1930,1960], (1960,1990], and (1990,2020], radial stem growth rate appears similar over the first ten years of growth. Over 20 years the most recent cohort shows marked acceleration compared to other cohorts. Sample sizes given as n, the number of cores in each cohort. Small circles are individual BAI for individual cores by cambial age. Curves are loess fits with 95% CI. For comparison, mean July temperatures for Kotzebue by decade were 1930-1960: 11.6 °C, 1960-1990: 12.1 °C, and 1990-2020: 13.0 °C.

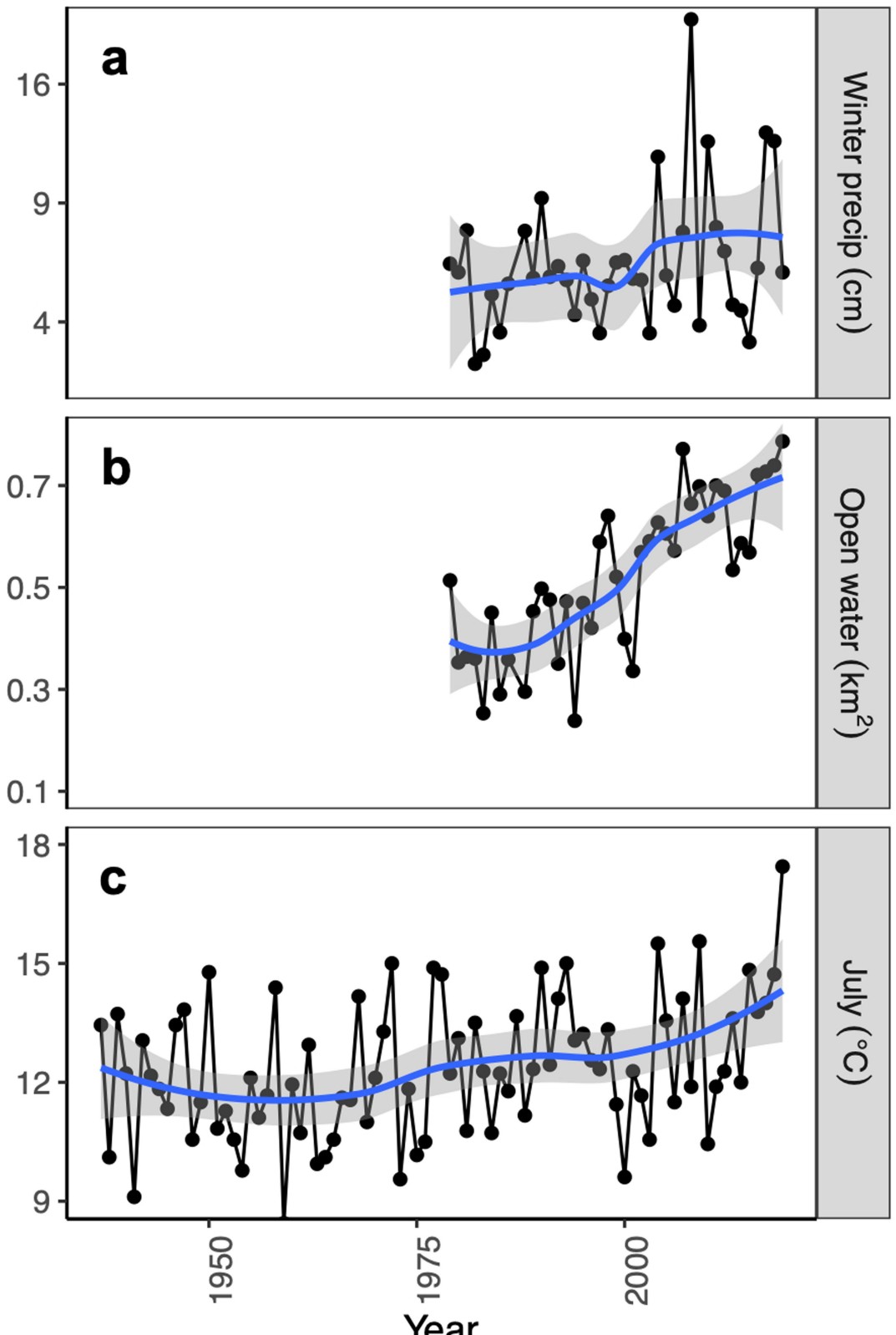

**Extended Data Fig. 6 | Air temperature, open water, and winter precipitation correlate in northwest Arctic of Alaska. a** Sum of December through March precipitation (square-root transformed) in Kotzebue, AK 1979-2019. Pearson's correlation with October open water in Chukchi Sea: $r = 0.41$ ($t = 2.73$, $df = 38$, $p = 0.009$). **b** Extent of October open water in Chukchi Sea 1979-2019. Pearson's correlation with July Kotzebue temperature: $r = 0.37$ ($t = 2.45$, $df = 38$, $p = 0.02$). **c** Mean daily July air temperatures at Kotzebue airport. Blue curves are loess fits. Note that precipitation scale is square-root transformed.

**Extended Data Table 1 | Upper Noatak basin imagery from https://evwhs.digitalglobe.com/**

| Date | Satellite platform | Image type | Solar Elevation (°) | Solar Azimuth (°) | Off- Nadir (°) | Extent S-N (°) | E-W (°) |
|------|--------------------|------------|---------------------|-------------------|----------------|----------------|---------|
| 26/3/2018 | WV1[*] | pan | 18 | 226 | 23 | 67.399 | 157.82 |
| | | | | | | 67.621 | 158.13 |
| 26/3/2018 | WV1[*] | pan | 18 | 226 | 23 | 67.421 | 157.64 |
| | | | | | | 67.613 | 157.9 |
| 1/4/18 | WV2[†] | pan-sharp color | 27 | 181 | 5 | 67.312 | 157.42 |
| | | | | | | 67.727 | 157.81 |
| 1/4/18 | WV2[†] | pan-sharp color | 27 | 181 | 5 | 67.48 | 156.92 |
| | | | | | | 68.018 | 157.37 |
| 4/4/18 | WV1[*] | pan | 22 | 226 | 21 | 67.479 | 157.17 |
| | | | | | | 67.593 | 157.61 |

*0.56m resolution †0.47m resolution.

**Extended Data Table 2 | Field sites used to estimate juvenile relative growth rate (RGR)**

| Field Site | Population | Basin | Watershed | Elevation (m asl*) | Samples | Lon, Lat (degrees) |
|---|---|---|---|---|---|---|
| AmakW | Colonist | Noatak | Amakomanak | 475-555 | 195 | 157.46, 67.54 |
| AmakE | Colonist | Noatak | Amakomanak | 460-510 | 81 | 157.36, 67.54 |
| AC Ridge | Colonist | Noatak | Cutler | 740-790 | 19 | 157.52, 67.53 |
| Cutler | Colonist | Noatak | Cutler | 305-515 | 7 | 158.17, 67.73 |
| PhilipSmith | Established | Noatak | Noatak | 130 | 60 | 161.52, 67.99 |
| AM upper | Established | Kobuk | Ambler | 210-620 | 15 | 156.81, 67.49 |
| AM lower | Established | Kobuk | Ambler | 190-200 | 14 | 156.84, 67.48 |
| KU lower | Established | Noatak | Kugururok | 150-190 | 13 | 161.75, 68.09 |
| KU upper | Established | Noatak | Kugururok | 180-260 | 12 | 161.77, 68.08 |
| KE upper | Established | Noatak | Kelly | 120-220 | 12 | 162.39, 68.08 |
| KE lower | Established | Noatak | Kelly | 190-260 | 12 | 162.32, 68.08 |
| AG upper | Established | Noatak | Agashashok | 141-151 | 2 | 162.21, 67.46 |
| AG lower | Established | Noatak | Agashashok | 93-95 | 2 | 162.22, 67.46 |
| TA upper | Established | Kobuk | Squirrel | 240-270 | 12 | 161.30, 67.45 |
| TA lower | Established | Kobuk | Squirrel | 170 | 12 | 161.31, 67.44 |
| TI lower | Established | Kobuk | Timber | 200-210 | 12 | 160.42, 67.37 |
| OM upper | Established | Kobuk | Omar | 260-290 | 11 | 160.66, 67.37 |
| TI upper | Established | Kobuk | Timber | 300-330 | 9 | 160.43, 67.38 |
| OM lower | Established | Kobuk | Omar | 150-180 | 6 | 160.65, 67.37 |
| AL upper | Established | Koyukuk | Alatna | 260-830 | 5 | 153.89, 67.47 |
| HF upper | Established | Koyukuk | John | 645-710 | 3 | 152.42, 67.81 |
| HF lower | Established | Koyukuk | John | 380-386 | 5 | 152.40, 67.79 |
| CL upper | Established | Koyukuk | N. Fork Koyukuk | 790-815 | 3 | 150.52, 67.77 |
| CL lower | Established | Koyukuk | N. Fork Koyukuk | 710-725 | 2 | 150.53, 67.77 |

* m asl=meters above sea level.

**Extended Data Table 3 | Foliage samples from established-treelines with on-going experiments (BRTL)**

| Basin | Watershed | Sample size |
|---|---|---|
| Noatak | Kelly | 30 |
| Noatak | Kugururok | 30 |
| Noatak | Agashashok | 30 |
| Noatak | Eli | 10 |
| Kobuk | Omar | 30 |
| Kobuk | Timber | 10 |
| Kobuk | Salmon | 10 |
| Kobuk | Ambler | 25 |
| Kobuk | Squirrel | 10 |
| Koyukuk | Alatna | 30 |
| Koyukuk | John | 30 |
| Koyukuk | Koyukuk | 10 |

# Reporting Summary

## Statistics

For all statistical analyses, confirm that the following items are present in the figure legend, table legend, main text, or Methods section.

| n/a | Confirmed | |
|---|---|---|
| ☐ | ☒ | The exact sample size ($n$) for each experimental group/condition, given as a discrete number and unit of measurement |
| ☐ | ☒ | A statement on whether measurements were taken from distinct samples or whether the same sample was measured repeatedly |
| ☐ | ☒ | The statistical test(s) used AND whether they are one- or two-sided<br>*Only common tests should be described solely by name; describe more complex techniques in the Methods section.* |
| ☐ | ☒ | A description of all covariates tested |
| ☐ | ☒ | A description of any assumptions or corrections, such as tests of normality and adjustment for multiple comparisons |
| ☐ | ☒ | A full description of the statistical parameters including central tendency (e.g. means) or other basic estimates (e.g. regression coefficient) AND variation (e.g. standard deviation) or associated estimates of uncertainty (e.g. confidence intervals) |
| ☐ | ☒ | For null hypothesis testing, the test statistic (e.g. $F$, $t$, $r$) with confidence intervals, effect sizes, degrees of freedom and $P$ value noted<br>*Give P values as exact values whenever suitable.* |
| ☒ | ☐ | For Bayesian analysis, information on the choice of priors and Markov chain Monte Carlo settings |
| ☐ | ☒ | For hierarchical and complex designs, identification of the appropriate level for tests and full reporting of outcomes |
| ☐ | ☒ | Estimates of effect sizes (e.g. Cohen's $d$, Pearson's $r$), indicating how they were calculated |

*Our web collection on statistics for biologists contains articles on many of the points above.*

## Software and code

Policy information about availability of computer code

Data collection
: Data collected in the field and recorded on smart phone apps for recording data on iPhone 12 Pro and Pro Max: We used Gaia GPS (https://www.gaiagps.com/) for locations which uses the phone's GPS; the native camera app for taking photos; we measured tree heights with an LTI laser range finder (https://lasertech.com/trupulse-series/) and with the Arboreal app (https://www.arboreal.se/en/) which uses the structure from motion from the phone camera and the iPhone 12 Pro's lidar; we entered data into the native iPhone app Numbers and a third party app Form Connect Pro (https://www.formconnections.com/formconnect-pro-form-designer/). The laboratory data we collected and the on-line sources of the climate, digital elevation model, sea ice, and imagery data are described in the methods. For example we used CooRecorder (https://www.cybis.se/forfun/dendro/index.htm) to scan increment cores and we imported satellite images into Google Earth Pro and digitized the shadows of trees there.

Data analysis
: We used the open source statistical software R in its latest version, R 4.1.2 "Bird Hippie" released on 2021/11/01. Our Supplementary Information provides the key code annotated and run as a pdf Markdown from RStudio. We tried to provide a readable walkthrough of the analysis for reviewers and interested readers, when/if the paper is published. All the R packages are listed below.

Other software is described in the methods in the appropriate locations: like Google Earth Pro, and two dendro-packages CooRecorder (https://www.cybis.se/forfun/dendro/index.htm) and CDendro (https://www.cybis.se/forfun/dendro/index.htm) software.

R packages (latest versions as of submission: January 18 2022) used:
bookdown
broom
Cairo
cowplot
data.table
doBy
dplR

```
emmeans
fields
ggridges
github("zeehio/facetscales")
Hmisc
lme4
MASS
nlme
plyr
raster
rcartocolor
reshape2
rgdal
rgeos
scales
sf
sp
stars
stringr
terra
tidyverse
viridis
```

For manuscripts utilizing custom algorithms or software that are central to the research but not yet described in published literature, software must be made available to editors and reviewers. We strongly encourage code deposition in a community repository (e.g. GitHub). See the Nature Portfolio <u>guidelines for submitting code & software</u> for further information.

## Data

Policy information about <u>availability of data</u>

All manuscripts must include a <u>data availability statement</u>. This statement should provide the following information, where applicable:

- Accession codes, unique identifiers, or web links for publicly available datasets
- A description of any restrictions on data availability
- For clinical datasets or third party data, please ensure that the statement adheres to our <u>policy</u>

Data and code used for analysis presented here are archived at the National Science Foundation's Arctic Data Center as doi:10.18739/A2X63B650.

# Field-specific reporting

Please select the one below that is the best fit for your research. If you are not sure, read the appropriate sections before making your selection.

☐ Life sciences   ☐ Behavioural & social sciences   ☒ Ecological, evolutionary & environmental sciences

For a reference copy of the document with all sections, see <u>nature.com/documents/nr-reporting-summary-flat.pdf</u>

# Ecological, evolutionary & environmental sciences study design

All studies must disclose on these points even when the disclosure is negative.

| Study description | This study compares attributes of a recently discovered disjunct population of a boreal conifer that colonized in the 20th century to attributes of established treelines, thereby describing for the first time sufficient conditions leading to boreal forest advance into Arctic tundra. These conditions are linked to a warming climate, increasingly open Arctic Ocean waters, and increased winter precipitation. |
|---|---|

Essentially all of the data presented is quantitative and covers five topics that tell the story:
1) The distance and rate of range expansion comparing adult to juvenile colonists that suggests acceleration of the range expansion.
2) The population growth of colonists from 1900-1980 equals a ten-year doubling time.
3) A comparison of individual growth metrics in colonists to individuals at established treelines (IET) shows that colonist growth is nearly double that of individuals in established populations.
4) A comparison of the environmental conditions between colonists and established treelines suggests that colonists occupy snowier, more nutrient-rich environments.
5) Regional climate variables over the historical period of instrumental recording show a century of average warming at over 2 degreeC/century since 1937 and near doubling of open water extent in the October Chukchi Sea and in winter precipitation over the last four decades

For 1) the dependent numeric variables include distances northward from established treelines (defined by 3,366 vertices of digitized lines) and elevations above (or below in the case of colonists beyond the mountain barrier) established treelines. Using the maximum age of trees sampled with increment cores another dependent variable is migration rate as distance divided by maximum age. The independent factor variable is age-class, with adults as trees over 2.5 m (n = 5,986), and juveniles (n = 770) as under 2.5 m. These data are presented within the framework of invasion theory and dispersal kernels and the context of documented recent spruce migration rates vs paleo-migration rates.

For 2) we measured 5,986 tree shadow lengths on satellite imagery and collected tree heights (n = 340 trees) and increment cores (n = 140 trees) from hundreds of trees in the field. We constructed Gaussian kernel density distributions by tree height class of n = 123 aged trees in five size classes: 3-4m (n = 31 aged trees with heights), 4-4.5 (n = 25), 4.5-5 (n = 21), 5-5.5 (n = 29), and 5.5-7m (n = 17)). We used linear mixed models with height as dependent variable, shadow length as independent variable and sample area as a random factor (random slopes and random intercepts) to construct a model to predict tree height based on shadow length for 1,971 (all shadows in two subwatersheds) of the 5,986 shadows repeatedly in a Monte Carlo method with 5,000 runs. For each run, we drew parameters from a multivariate normal distribution to estimate 1,972 tree heights based on shadow length. For each of the 1,971 trees we then drew an age from the appropriate kernel density of ages given the tree's height class. We aggregated by year and cumulatively summed to get population size by year from 1900 to 1980. For each of these 5,000 runs we fit an exponential model using a non-linear least squares algorithm. Averaging the exponential growth rate gave our best estimate of population growth of the 1,971 trees that cast the shadows in the two subwatersheds.

For 3) we used three growth metrics as dependent variables: (i) Current annual growth (CAG) in lateral branches of adults (n = 17 colonists from m = 6 watersheds and n = 457 IET from m = 12 watersheds) at 1.4 m above ground. (ii) Relative growth rate (RGR) in height during the last five years (2015-2020) in juveniles (n = 300 colonists from m = 4 field sites and n = 271 IET from m = 20 field sites)). And (iii) Radial stem growth in juvenile (n = 15 trees < 30 years old) and adult (n = 125 trees ≥ 30 years old) colonists as measured using increment cores and two standard ring indices, basal area index (BAI) and autoregressive residuals (AR). Each individual tree ring chronology was correlated with Pearson's correlation with July temperature from Kotzebue; no -values were calculated. For comparisons between colonist and established treeline populations, measured adults and juveniles were replicated within random factor "areas" as watersheds (CAG) or field sites (RGR). These areas served as random factors in linear mixed effects models. The independent fixed factor variable for (i) and (ii) was source population of the individual as "colonist" or "established treeline" populations. For RGR an included covariate was juvenile height, because RGR decreases with height. The linear mixed effects model included random intercepts; the interaction of the covariate juvenile height with source population provided the fixed effects. For CAG, there was no covariate, only the fixed factor of source population and the random intercept of watershed.

For 4) we compared two types of environmental variables: nutrient and climatic. We used three dependent nutrient variables: percent foliar phosphorous (n = 19 colonists at m = 6 watersheds; n = 51 pooled colonist needle samples from m = 14 watersheds), percent foliar nitrogen (n = 20 colonists at m = 6 watersheds; n = 51 pooled colonist needle samples from m = 14 watersheds), and the ratio of delta N15 to N. Each nutrient variable was treated as described above in 3) (i) for CAG. That is, we employed a watershed as a random factor for intercept and population ("colonist" or "established treeline") as fixed factor. We employed two gridded climate variables each with a biological basis: mean July temperature and winter precipitation. We performed no inferential statistics with these variables. Instead we graphically compare the bivariate kernel density distributions of the two variables for (i) the watersheds supporting the colonist populations, (ii) the nearby watersheds supporting established treelines, (iii) the gridded climate pixels enclosing the locations of the colonists (n = 708 pixels; if multiple individuals were found in one pixel the pixel is reported only once) and (iv) ) the gridded climate pixels enclosing the locations of established treeline vertices (n = 1,213 pixels; if multiple vertices were contained in one pixel the pixel is reported only once).

For 5) we report the instrumental records of July temperature and winter precipitation as available from the US Government's NOAA web site and use the sea ice extent in October to estimate the amount of open water in October over the period of reporting on the National Snow and Ice Data Center. Here we perform simple one-tailed Pearson correlation tests.

| | |
|---|---|
| Research sample | We sampled white spruce, Picea glauca, growing at and beyond treeline at the northwest extreme of their geographic range in Arctic Alaska, USA. The sample is meant to represent the population of white spruce that has recently expanded beyond the known range of the species. |
| Sampling strategy | We collected as many colonist samples as we could given the time and weather constraints of remote fly-in/walk and float out self-contained expeditionary field science. At the established treelines our samples were pre-determined by experimental design of 10 trees per study plot with one to three study plots per study site and two to six study sites per remote wilderness access site by fixed wing, single engine airplane. In addition, we searched for and recorded locations of colonists found beyond treeline in 37 other watersheds. |
| Data collection | RJD led four field campaigns to document, map, measure, and collect foliar and wood samples of white spruce colonists to the area of interest (AOI) and seven field campaigns outside the AOI to search for spruce colonists. He was accompanied by one to six field assistants. Field assistants Logan Berner and Patrick Burns of Northern Arizona University and Ray Koleser, formerly of the United States Forest Service Forest Inventory and Analysis program cored the trees. RJD, Julia Ditto, Ray Koleser, Brad Meiklejohn, Patrick Burns, Ben Weissenbach, Russell Wong, and Madeline Zietlow participated in the following: measuring heights above ground of bud scars of juvenile colonists with a tape measure and adults with a laser range finder and/or phone app; measuring DBH; mapping colonists with a GPS; photographing individual trees; recording ecologically relevant information such as cone abundance, growth form, damage, abundance of shrubs and juveniles within 5m of measured adults. Annie Brownlee and Jocyln Kramer assisted in laboratory processing and dendrochronology supervised by CTM and PFS. CTM measured rings and cross-dated increment cores. CTM, PFS, Julia Ditto, Ben Weissenbach, Madeline Zietlow, Russell Wong, measured bud scar heights of juveniles at established treelines. RJD and Sylvia Taylor digitized shadows on satellite imagery. RJD, CTM, PFS, Russell Wong and Scout Donahue collected foliar samples. REH and PFS supervised foliar nutrient data collection. RJD downloaded on-line data including Digital Globe World-View imagery, PRISM gridded climatology, IFSAR Digital Elevation models, Kotzebue and RAWS climate data, and Chukchi Sea Ice area. |
| Timing and spatial scale | We collected data from the colonist population within the AOI in August and September of 2019; September 2020, and July 2021. We searched for and found spruce colonists outside the AOU in June and August 2018, June 2019, July and August 2020, and June-August 2021. We collected data from established treelines in August-September 2019, and June-September 2021. The spatial extent of the data collection covers 1000 km east-west and 200 km north-south approximately. Most of the colonist AOI data were collected in an area 90 km east-west and 55 km north-south. |
| Data exclusions | We exclude data only from the population simulation. In that case we excluded trees less than 3 m tall because they were unlikely visible in the remote sensing imagery, buried by snow and casting short shadows. We also exclude two locations of spruce beyond treeline. These were the only two krummholz tree islands that we encountered distant from treeline and we did not consider them |

to be colonists. We also discarded 10 of 150 tree cores because they lacked height above ground measurements where increment cores were taken.

| | |
|---|---|
| Reproducibility | We collected data during 12 field campaigns, where a field campaign is defined by a discrete field expedition to the Brooks Range led by one of the co-authors. Because the study areas are distant and expensive to access, our goal was to visit as broad an area as possible. We did not repeat any measurements in the AOI but we used the same sampling methods we used on previous trips plus generally new ones as well as we learned more about the area and new questions arose. This study was more akin to scientific field exploration than a replicated experiment. However we have ongoing experiments at established treelines that are included in this study for comparison, but we include current annual growth measurements (CAG) from only 2019 (since we only measured CAG from the AOI that year) and relative growth rates (RGR) are from 2021 at our experimental study sites because we only measured RGR in the AOI in 2021. |
| Randomization | Field sites among the established treelines were selected randomly conditional on being at treeline and within 4 km of the airstrip access. For the colonist populations or transects were not random, but chosen based on remote sensing imagery and the ability to travel on the terrain. Four field campaigns (2019-2021) focused on three objectives in watersheds that were within or adjacent to the Noatak River basin but without established treelines visible on WV growing-season scenes: (1) locate and document colonists at the geographic range boundary of white spruce, (2) verify locations of a sample of trees suggested by imagery in the AOI, and (3) collect ecological measurements germane to white spruce range expansion. We selected representative (i.e. stratified random sampling by height class) and extreme individual adults (i.e., we sought out the highest, tallest, farthest north, and biggest and oldest looking individuals). An additional five field campaigns (June and August 2018, June 2019, July-August 2020, and June-August 2021) were undertaken to locate and record whiet spruce beyond established treelines during very long sample transects that were not random, but instead selected to explore treeline conditional on ability to travel and be resupplied by small fixed-wing, single-engine aircraft. |
| Blinding | For field data collection blinding is impractical. |

Did the study involve field work?  ☒ Yes  ☐ No

# Field work, collection and transport

| | |
|---|---|
| Field conditions | The Brooks Range extends from the Canadian Border to the Chukchi Sea, roughly 1000 km east-west. In the east it is dry and very cold in winter. In the center it receives more precipitation. In the west it is warmest and wettest. Vegetaion along treeline is mostly dominated by white spruce with an understory of willow, birch, and occasionally alder. The ground cover included ericaceous and other dwarf shrubs, especially Dryas, growing among gramminoids, mosses, and lichens.<br><br>Only one road crosses the Range roughly one-third of the distance from Canada to Kotzebue. Within the Range itself there are only three communities, two on the road. Nearly all of the Brooks Range is protected in Federally designated conservation areas that protect the Range from extractive activities including mining and logging. At treeline forest fires are uncommon although thawing permafrost is leading to increased disturbance by frozen debris lobes.<br><br>Northwest Alaska's Noatak and Kobuk Rivers are located above the Arctic Circle and drain into the Chukchi Sea of the Arctic Ocean near Kotzebue (162.604°W, 66.891°N). The Baird Mountains of the southwestern Brooks Range separate the Kobuk from the Noatak basins and the De Long Mountains of the northwestern Brooks Range separate the Noatak from North Slope river basins and from the Wulik basin. The lower basins of the Noatak and Kobuk Rivers are included in Alaska's West Coast climate division, characterized by greater precipitation, warmer winters, and cooler summers than the Central Interior climate, with greater precipitation and warmer temperatures than the North Slope climate. The upper basin of the 700 km Noatak lies at the intersection of all three climate divisions, which warmed 1949-2012. December-January precipitation increased 1949-2012 in the West Coast climate division, as did North Slope winter precipitation 1980-2012.<br><br>Vegetation includes dwarf, low, or tall shrub tundras that cover about 60% of the 33,000 km2 Noatak River basin (entirely protected within federal conservation units), with tussock sedge tundra covering another 30% and wetlands and barrens most of the remainder The main valley and tributaries along the lowest 200 km of the Noatak River support stands of white spruce, typically associated with greater depth or absence of permafrost. The treelines bounding these forests have long been identified as the northwest range extent of white spruce.<br><br>The upper Noatak basin, a 500 km reach, is underlain by extensive continuous permafrost. It has been considered empty of spruce since USGS geologist Philip Smith explored the Kobuk, Alatna, and Noatak Rivers by canoe in 1911. The adjacent Kobuk and Alatna River basins support boreal forests of black and white spruce, paper birch, and aspen along much of their lengths. By surveying transects along and beyond hydrological divides separating the Noatak, Wulik, Kobuk, and Alatna River basins, informed by very-high resolution satellite scenes, we documented the locations of over 7,000 individual spruce colonists. |
| Location | The location of the AOI includes four watersheds of the Cutler River a tributary of the upper Noatak River in northwest Alaska's Brooks Range and four adjacent watersheds in the upper Kobuk River basin. The center of the eight watersheds is roughly located at 157.7 W 67.5 N. The watersheds include the Kaluich, Cutler, Amakomanak, and Imelyak Rivers in the Noatak Basin and the Akillik, Redstone, and Redstone, Miluet, Akillik, and Hunt watersheds (HUC 10) of the Middle Kobuk subbasin as defined by the US Governments USGS "Hydrologic Unit Classification" system. In addition, we cite data from 37 other watersheds that are spread across the Range. Here are there locations: Philip Smith North Fork East Fork Chandalar River 147.424851 -147.424851<br>Philip Smith Sheenjek River 143.612821 -143.612821<br>Endicott Clear River 150.69185 -150.69185<br>Endicott North Fork Koyukuk River 150.69901 -150.69901<br>Endicott Iniakuk River 153.04305 -153.04305<br>Endicott Wolverine Creek 152.90086 -152.90086<br>Endicott Kutuk River 153.706 -153.706<br>Endicott Lucky Six 154.859 -154.859<br>Schwatka Ipnelivik 156.294 -156.294 |

```
Schwatka Reed 155.061 -155.061
Baird Timber Creek 160.420162 -160.420162
Philip Smith Wind River 147.148 -147.148
Endicott Publituk Creek 151.90186 -151.90186
Endicott Wild River 151.67801 -151.67801
Endicott Agiak Creek 152.921 -152.921
Baird Omar River 160.518988 -160.518988
Baird Squirrel 161.456217 -161.456217
De Long Wulik 163.22 -163.22
Philip Smith Middle Fork Chandalar River 147.523 -147.523
Endicott Tinayguk River 151.48682 -151.48682
Endicott Pingaluk River 153.46783 -153.46783
Schwatka Kugrak 155.723 -155.723
Baird North Fork Squirrel 161.241982 -161.241982
Endicott Glacier River 150.40739 -150.40739
Endicott Nahtuk River 153.17322 -153.17322
Endicott Akabluak east 154.461 -154.461
Endicott Awlinyak 154.342 -154.342
De Long Upper Wrench 162.624 -162.624
Endicott Kevuk Creek 152.68016 -152.68016
Schwatka Imelyak 157.004 -157.004
Endicott Wolf Creek 151.29355 -151.29355
Endicott Akabluak west 154.777 -154.777
Endicott Arrigetch Valley 154.185 -154.185
De Long Wrench Ck south 162.544 -162.544
Philip Smith W. Sheenjek River 147.424851 -147.424851
De Long Wrench Ck woods 162.596537 -162.596537
Philip Smith North Fork Chandalar River 149.406395 -149.406395
Chukchi Sea Cape Krusenstern 67.040703 -163.11433
Noatak Noatak. 158.34332 -158.34332
Noatak Noatak.158.22879 -158.22879
```

| | |
|---|---|
| Access & import/export | We applied for and received the following United States National Park Service Permits to conduct research in the Gates of the Arctic National Park and Noatak National Preserve conservation units: # NOAT-2021-SCI-0002, # GAAR-2021-SCI-0004, # GAAR-2019-SCI-0002. In order to reduce our environmental impact we used aircraft only to fly-in to the colonist population. from there we walked and rafted out to small villages. |
| Disturbance | We camped following "leave no trace principals" and made every effort not to disturb the wilderness environment or its wild inhabitants. |

# Reporting for specific materials, systems and methods

We require information from authors about some types of materials, experimental systems and methods used in many studies. Here, indicate whether each material, system or method listed is relevant to your study. If you are not sure if a list item applies to your research, read the appropriate section before selecting a response.

## Materials & experimental systems

| n/a | Involved in the study |
|---|---|
| ☒ ☐ | Antibodies |
| ☒ ☐ | Eukaryotic cell lines |
| ☒ ☐ | Palaeontology and archaeology |
| ☒ ☐ | Animals and other organisms |
| ☒ ☐ | Human research participants |
| ☒ ☐ | Clinical data |
| ☒ ☐ | Dual use research of concern |

## Methods

| n/a | Involved in the study |
|---|---|
| ☒ ☐ | ChIP-seq |
| ☒ ☐ | Flow cytometry |
| ☒ ☐ | MRI-based neuroimaging |

