## [Peer Review File · Nature]

Manuscript Title: Sufficient conditions for rapid range expansion of a boreal conifer

Reviewer Comments & Author Rebuttals

Reviewer Reports on the Initial Version:

Referees' comments:

Referee #1 (Remarks to the Author):

The Editor asked me to focus on the dendrochronological component of this study, and so I will direct most of my comments to that part of the paper.

The authors obtained tree-ring width data from nearly 150 trees in their sample and use those measurements to estimate (1) trends in growth during the past three decades and (2) to argue that summer temperature is the main environmental factor that has made growth at this site so rapid.

To estimate growth trends, the authors use tree-ring width measurements to compute basal area increment (BAI) series. Switching over to BAI is sometimes presented as an effective means to sidestep the strong age-size trends that are evident in tree-ring width series. I do worry that BAI calculations also bring with them their own assumptions — mainly, that trees maintain a perfectly circular cross-section during their lifetime, which is usually not true for long lived trees. But because the trees in this study are all relatively young, I am not too concerned about the risks that the BAI computation itself could introduce spurious trends.

However, I would very much like to see what growth looks like at this site over the full period of record — not just the most recent 30 years. Because the authors argue that tree growth at this site is now progressing at an unprecedentedly high rates, I think we need to see growth rates prior to 1990. Either in the main manuscript or supplemental information, I would want an extended version of Figure 3C. It should also be possible to extend the yellow line (which represents the growth of juvenile trees) by including the juvenile portion of growth (the first 30 years) for the now-older trees. I understand that the uncertainty of the mean plots will increase back in time due to the diminishing number of trees included in the average. But I think this slightly longer perspective would strengthen the authors argument that modern growth rates (since 1990) are really unusual.

Also, it's striking to me that these trees are growing so much faster now, compared to 1990. A ten-fold increase in BAI is huge, and is so big that the same pattern must be visible in the raw tree-ring measurements. And so I would ask the authors to also plot the tree-ring width series for their ~150 trees. Notwithstanding the inherent trend towards more narrow rings as trees age, if the growth increase is as large as indicated by the BAI plots, I would expect to see an upward trend in the tree-ring width measurements as well.

On the second topic — the influence of climate on tree growth — I agree there are some situations where it's necessary to remove memory/persistence from tree-ring series. But in this specific case, it

seems to me the authors are assuming that all memory in tree-ring widths is biological in origin, and so they remove that element to make a cleaner comparison between tree growth and climate. But the climate also often exhibits strong memory or persistence (see <https://www.nature.com/articles/s41598-018-33217-0> for a recent article on temporal autocorrelation in temperature). I don't think it's appropriate to strip away persistence from one variable (tree growth) and not apply the same correction to the other (July temperature). There is a useful schematic in Coulthard et al. (2020; <https://www.sciencedirect.com/science/article/abs/pii/S0277379120300597?via=ihub>) that reminds us that memory is an important component in both the physical and biological environment). It's also striking that the step to remove memory seems to (substantially) strengthen the apparent temperature signal in tree growth. In Figure 3.d. in the Supplemental Information, the mean correlation between the two variables (for adult trees) shifts from ~ 0.2 to ~ 0.37 . I'm not sure this is a problem, but I would feel more confident in this result if it was shown to be insensitive to the asymmetrical application of prewhitening (in other words, do you get the same result of you prewhiten tree growth AND temperature?).

The Editor also asked me to consider how these findings could be generalized or compared against other species or sites. I'm not aware of other tree-ring studies that have shown enhanced growth of white spruce over the past few decades. Most similar collections in Alaska or the Yukon were now conducted 10 or 20 years ago, so those data would not contain information about the main time period considered by this study (1990 to 2019). If the authors can demonstrate that recent growth trends are truly exceptional (by extending their time series plots prior to 1990), that could be an important result that gives the rest of us something to hunt in other locations.

Referee #2 (Remarks to the Author):

This study gives evidence that boreal conifers have expanded their populations into new areas, which is very interesting and the evidence using both satellite and field data seems to be compelling. Satellite data are mainly used to scale field observations to larger areas, and while the method is nothing special, it seems to work for this case. The study uses WorldView images, both WorldView-1 and WorldView-2, to manually count specific tree species which are only visible in winter, as they clearly stand out from the snow. The only evidence on the applicability of this approach is Figure 1, but it is not clear to me why the shapes under the yellow ovals are trees but not those on the left side, I really cannot see a difference. It is also a bit unclear to me if the study area is the shaded area from the small map.

Artificial intelligence may be used in future studies to automate the detection process over much larger areas, where a manual counting is not feasible. Figure 1 also shows a color image, which is cited as "DigitalGlobe Vivid", please make sure this is fine to show concerning copyrights. Figure 1 b says "revealing five trees", but I can only see four yellow ovals.

The method on the remote sensing part may be slightly improved and more details on many trees were counted, how this was validated, etc may be provided. Was it a wall-to-wall mapping or were only the clear ones mapped? Also, I was looking for a map illustrating the results from the satellite

image based mapping, but i only found ED Figure 2, which seems to be a small subset of the studied area, but it's also not clear if this comes from field work or satellite images. I assume ED Figure 1 shows the counts from the field work?

Referee #3 (Remarks to the Author):

The authors have identified a previously unknown population of young spruce trees in Alaska, well beyond the tree line and expanding rapidly, possibly at an unprecedented rate. I am not enough of an ecologist to say whether this is in fact so, but I think well-founded research that bears upon the issue of the mismatch between climate change velocity and species advance rates in the subarctic is important. These rates differ by 2-3 orders of magnitude, so that the distribution of northern vegetation is increasingly out of equilibrium with the climate, and the question of what form approaches towards equilibrium will take is of great interest. The authors conclude their main text by suggesting that coniferous trees may be on the verge of invading the subarctic tundra. Perhaps this is a step beyond what can be supported by the research described here, but I am quite excited by this work. I see it as primarily phenomenological research (I am less interested in the attempts to link the results to specific environmental factors in the study region), with a strong component of methodological development, and I would like to see it published to encourage more studies like this to be undertaken.

Technical issues

I have focussed, as requested (and as fits my expertise), on the remote sensing aspect of this study. Here, the authors have used very high resolution satellite remote sensing data as a means to estimate the height, and hence age class, of trees, together with their locations. The process is in general well described in the manuscript and in the portfolio summary. However, it does raise some questions for me. I think these will probably all have satisfactory answers, but here they are:

1. The ability to match trees measured in situ with those identified in the WV imagery depends on the georeferencing accuracy of both the imagery and the GPS used for survey, as well as the spatial distribution of trees. The authors don't give many details and it would be good to know how confident they are the matches are valid. In fact they report in the 'portfolio' section that field locations were obtained using iphones of various kinds, so (to be pedantic) these will have made use of several other GNSS systems as well as GPS, and I would expect the accuracy to be something like 3 m unless within dense forest. Were the WV images acquired close to nadir (i.e. easier to create orthoimages) or far from it?
2. Almost 6000 shadow lengths were digitised by five individuals. Did all five measure all the shadows, or did they do non-overlapping samples? If so, did they make any attempt to identify and if necessary correct for systematic differences between them?
3. Height classes deduced from the shadow measurements were in some cases only 0.5 m in width. If the mean snow depth (which has to be added to the shadow-deduced height) differed by only this much from one part of the study area to another, this would introduce a systematic misclassification

between locations. How confident are the authors that this is not an important effect?

Referee #4 (Remarks to the Author):

I have reviewed with great interest the manuscript by Dial et al. "Sufficient conditions for climate-driven range expansion of a boreal conifer". Overall, I think this is an important, complete and well-written piece that shows undisputable evidence that the forest is advancing, benefiting, under certain circumstances, from global warming. Ecological processes, like seed dispersion and other population-level ones, have been proposed to explain this vegetation advance. My only major concern with the present study is the general idiosyncratic scope of it. Is *Picea glauca* colonizing other environmentally similar spots in the boreal region? Literature from Canada is missing in this respect; probably, there is no literature documenting similar phenomena, but this argument is not included in the Discussion. I think it is important to bring some invasion ecology theory into the pattern studied but more importantly would be to scale up inference and show the potential for other areas in the boreal region to sustain similar vegetation advances. For example, in the abstract it says: "This species range expansion, cast in the context of invasion theory, informs forecast models of vegetation change with conditions driving biome shift from tundra to forest" and here it is where I would have expected to have the actual forecast including those vegetation models. High soil nutrient availability is described as a factor that could explain the establishment of forest in the new location; do you think a higher soil nutrient availability is a consequence of global warming? If so, in which way? In general, my concern centers on how peculiar the environmental conditions of the colonized area are. In other words, how generalizable these results are?

Author Rebuttals to Initial Comments:

Response to Referees:

Nature manuscript 2022-01-00658

Note from authors to Editor and Referees:

We have provided both a “track-changes” and a “clean copy” version. Please note that line references in our responses apply to the clean copy.

Referee #1 (Remarks to the Author):

Referee #1 comments “The Editor asked me to focus on the dendrochronological component of this study, and so I will direct most of my comments to that part of the paper.

“The authors obtained tree-ring width data from nearly 150 trees in their sample and use those measurements to estimate (1) trends in growth during the past three decades and (2) to argue that summer temperature is the main environmental factor that has made growth at this site so rapid.

“To estimate growth trends, the authors use tree-ring width measurements to compute basal area increment (BAI) series. Switching over to BAI is sometimes presented as an effective means to side step the strong age-size trends that are evident in tree-ring width series. I do worry that BAI calculations also bring with them their own assumptions — mainly, that trees maintain a perfectly circulator cross-section during their lifetime, which is usually not true for long lived trees. But because the trees in this study are all relatively young, I am not too concerned about the risks that the BAI computation itself could introduce spurious trends.”

AUTHORS' RESPONSE We appreciate this comment regarding BAI and its limitations. We chose BAI because ring area more closely represents actual radial growth than raw ring width, because BAI accounts, in part, for the size/age trend that Referee #1 describes. BAI also represents a minimalist approach to tree-ring data processing that avoids potential loss of information and/or

introduction of bias known to affect common methods such as negative exponential detrending.

As noted by Referee #1, BAI calculations have their own assumptions, perhaps the primary one being circular geometry of the bole's cross-section. The general youthfulness of individuals (all <120 years old), their symmetric crowns, and the relatively low population density of this rapidly "migrating" population would appear to meet the circular cross-section requirement. The symmetry of the cross-sections is highlighted by the fact that 71% of the 140 increment cores either contained the pith or missed it by five or fewer years.

We mention these conditions in Methods on

Lines 940-942: We selected trees with generally symmetric crowns in open locations with bark encircling the entirety of the mostly circular bole where we cored.

Referee 1 continues "However, I would very much like to see what growth looks like at this site over the full period of record — not just the most recent 30 years. Because the authors argue that tree growth at this site is now progressing at an unprecedentedly high rates, I think we need to see growth rates prior to 1990. Either in the main manuscript or supplemental information, I would want an extended version of Figure 3C. I understand that the uncertainty of the mean plots will increase back in time due to the diminishing number of trees included in the average. But I think this slightly longer perspective would strengthen the authors argument that modern growth rates (since 1990) are really unusual."

AUTHORS' RESPONSE Referee #1 asks to extend ring indices over the full chronology. In response we have extended the temporal range of the former Main Text Figure 3c (now **Fig. 3d**) to include all years with a sample size of at least 5 trees. The chronology now goes back to 1952. As anticipated by Referee #1, this extended ring series highlights the recent and unusually rapid growth rates. To make this dramatic result more accessible we have produced an additional Extended Data figure. The panels of the new **ED Fig 4** display a photographic scan (**ED Fig 4a**) of 21 cores that include their piths alongside a traditional dendrochronology spaghetti plot (**ED Fig 4b**). Together, these two

panels show the raw ring width data of all cores alongside an example of the wood from which the data derive.

The revised main text includes the following on

Lines 166-167: The exponential growth of the most recent decades contrasts with earlier growth reaching back to the 1950s (Fig. 3d).

Referee 1 comments “It should also be possible to extend the yellow line (which represents the growth of juvenile trees) by including the juvenile portion of growth (the first 30 years) for the now-older trees.”

AUTHORS' RESPONSE We welcome this suggestion to compare early growth in older trees during their first 30 years to the modern growth in current juveniles. We provide four plots to examine this comparison in the revised **SI 7.8**. Each plot displays the first 30 years of growth for all trees against their cambial age (years from pith) or against calendar years. Two of these plots shows four cohorts of juveniles from: 1900-1930, 1930-1960, 1960-1990, and 1990-2020. One of these cohort plots is a new **ED Fig. 5** which plots BAI against cambial age to compare the change in growth by cambial age over the decades. It makes clear that after about age 10, the increased rate of juvenile growth in the most recent cohort during the warmest decades is more rapid than the other cohorts during their first 30 years.

New main text in the Results includes changes on

Lines 172-176: When considering only cambial ages <30 y (thereby comparing growth rates of all trees when juvenile) juvenile growth during the last thirty years out-paced juvenile growth in previous decades (ED Fig 5, SI 7.9), suggesting an improvement in growing conditions coincident with recent rapid warming.

The revised Discussion includes the following on

Lines 232-234: The tree-ring growth chronologies as RRW and BAI showed exponential individual growth over the last three decades, with radial stem growth of modern juveniles exceeding the growth rates of previous juvenile cohorts.

Referee 1 comments “Also, it’s striking to me that these trees are growing so much faster now, compared to 1990. A ten-fold increase in BAI is huge, and is so big that the same pattern must be visible in the raw tree-ring measurements. And so I would ask the authors to also plot the tree-ring width series for their ~150 trees. Notwithstanding the inherent trend towards more narrow rings as trees age, if the growth increase is as large as indicated by the BAI plots, I would expect to see an upward trend in the tree-ring width measurements as well.”

AUTHORS’ RESPONSE We have added a raw ring width (RRW) chronology as an additional panel in the revised Main Text **Fig. 3d**. The new panel for RRW shows the upward trend visible in the other two indices. In addition, and as mentioned above, we have added a new **ED Fig. 4** that places a photographic scan of all cores that included pith (n = 21) adjacent to a classic dendrochronology “spaghetti plot” showing all tree ring series (code in **SI 7.3**). We feel that **ED Fig. 4a**--where rings are visibly narrow near the pith but show clear recent widening across all ages of trees as indicated by length of core and total number of rings--will appeal to a general, non-specialist reader. We hope that **ED Fig. 4b** will convey important information to those like Referee #1 who would like to see what is, in essence, the raw growth data ranked by length of each tree’s ring series.

As noted above, the new main text includes the following on

Lines 164-167: The radial growth of colonists was exponential from 1989-2019 in both (Fig. 3d) growth indices, RRW (ED Fig. 4) and BAI (SI 7). The exponential growth of the most recent decades contrasts with earlier growth reaching back to the 1950s (Fig. 3d).

And on

Lines 232-234: The tree-ring growth chronologies as RRW and BAI showed exponential individual growth over the last three decades, with radial stem growth of modern juveniles exceeding the growth rates of previous juvenile cohorts.

Referee #1 comments “On the second topic — the influence of climate on tree growth — I agree there are some situations where it’s necessary to remove memory/persistence from tree-ring series. But in this specific case, it seems to me the authors are assuming that all memory in tree-ring widths is biological in origin, and so they remove that element to make a cleaner comparison between tree growth and climate. But the climate also often exhibits strong memory or persistence (see <https://www.nature.com/articles/s41598-018-33217-0> for a recent article on temporal autocorrelation in temperature). I don’t think it’s appropriate to strip away persistence from one variable (tree growth) and not apply the same correction to the other (July temperature). There is a useful schematic in Coulthard et al. (2020; <https://www.sciencedirect.com/science/article/abs/pii/S0277379120300597?via=ihub>) that reminds us that memory is an important component in both the physical and biological environment).

AUTHORS’ RESPONSE Referee #1 points out that strong climate trends can lead to autocorrelated time series. To address this possibility, we applied the same AR “pre-whitening” algorithm to the annual July mean temperature series used for each tree-ring series. This algorithm from the `dplr` package in R returns residuals from an autoregressive model with the best-fit order selected by AIC. This generated an AR model with order = 0, indicating that the raw series already approximated white noise. The `dplr` package performs an AR algorithm from the “base” R package called `AR()` whose default is the Yule-Walker method.

To verify the `dplr` package result, we repeated the AR modeling process using the “base” R package function `ar()`, but with the Maximum Likelihood Estimation (`mle`) option instead of Yule-Walker. We again found the best order was zero.

We further investigated this result by plotting the partial autocorrelation function for orders 0-19, where it was apparent that the strongest autocorrelation occurred at a lag (*i.e.*, order) of 10 years. Plotting AIC as a function of order (shown in **SI**

7.2) demonstrates that the partial correlation at order = lag = 10 was insufficient to outweigh the additional complexity of an AR(10) model.

The annotated code that develops these results is provided in **SI 7.2**, while the relevant text in the Methods section is at

Lines 969-972: We also applied the AR approach to the July mean temperature data, but two common methods of selecting AR models (Yule-Walker and maximum likelihood) indicated that these data already approximated white noise (AR order = 0; SI 7.2). Thus, we retained the raw temperature data in subsequent analyses.

There is also a statement added to Results at

Lines 157-158: The Kotzebue July air temperature time series showed no temporal autocorrelation (SI 7.2) and so required no pre-whitening.

Referee 1 comments “It’s also striking that the step to remove memory seems to (substantially) strengthen the apparent temperature signal in tree growth. In Figure 3.d. in the Supplemental Information, the mean correlation between the two variables (for adult trees) shifts from ~0.2 to ~0.37. I’m not sure this is a problem, but I would feel more confident in this result if it was shown to be insensitive to the asymmetrical application of prewhitening (in other words, do you get the same result if you prewhiten tree growth AND temperature?).”

AUTHORS’ RESPONSE Referee #1 asks a good question: if we pre-whiten both temperature and tree-ring series do we get the same correlations between them? And the answer is yes, because the temperature data are apparently no different from white-noise, given the AR order of zero from two AR methods (Yule-Walker and MLE). Thus, correlations between tree-ring series and temperature, whether temperature is pre-whitened or not, are the same.

Referee 1 comments “The Editor also asked me to consider how these findings could be generalized or compared against other species or sites. I’m not aware of other tree-ring studies that have shown enhanced growth of white spruce over the past few decades. Most similar collections in Alaska or the Yukon were now conducted 10 or 20 years ago, so those data would not contain information about the main time period considered by this study (1990 to 2019). If the authors can demonstrate that recent growth trends are truly exceptional (by extending their time series plots prior to 1990), that could be an important result that gives the rest of us something to hunt in other locations.”

AUTHORS’ RESPONSE In response to the Editor’s query regarding generality and/or comparisons to other species or sites, Referee #1 responded that other studies in the Alaska/Yukon region are 10-20 years old and so likely miss most of the very rapid recent warming of the last three decades. Referee #1 goes on to further to point out the importance to other workers in the field of the recent upturn in growth rates.

The only other published Brooks Range chronologies currently known to exhibit such vigorous growth and such strong responses to warming occur in the western Brooks Range, on a riverside terrace that is well below treeline and known to have exceptionally warm soils (for the Arctic) with relatively high soil nutrient availability (Sullivan et al. 2015 (ref 31)). Most white spruce at established treelines in the Brooks Range have shown positive growth responses to warming during the first half of the 20th century. During recent decades, however, many of those trees have slowed in their growth response to rising air temperature (Wilmking et al. 2004 (ref 29)). The trees in the Cutler River basin (which is the biggest tributary of the Noatak and into which all four Arctic watersheds within the AOI drain) are unique relative to trees at established treelines, in the sense that nearly all individuals in the Cutler basin have increased their growth dramatically with warming over the last three decades. We suspect that other colonists across the Brooks Range will show this growth as well. We have learned from the Cutler that colonization of tundra by trees can occur at much further distances from established treelines than previously thought. We suspect that a deep winter snowpack is a key ingredient for tree invasion. These insights have allowed us to successfully predict the locations of vigorously growing juvenile spruce, including saplings, in many other locations well beyond established treelines in the Brooks Range. We predict that these more recently colonized areas will soon support self-sustaining populations of prolific mature trees, much like the Cutler.

Our new Discussion paragraph distills these observations as

Lines 239-244: However, because most tree-ring chronologies elsewhere in the Brooks Range are now one to two decades old and so do not include the most recent warming, it is difficult to know how extensive rapid radial growth might be. The only Brooks Range trees previously reported³¹ to exhibit the strong responses to recent warming seen here grow well below treeline in the lower Noatak basin on a riverside terrace with known warm soils and high soil nutrient availability.

Sullivan, P.F., Ellison, S.B., McNown, R.W., Brownlee, A.H. and Sveinbjörnsson, B., 2015. Evidence of soil nutrient availability as the proximate constraint on growth of treeline trees in northwest Alaska. *Ecology*, 96, 716-727.

Wilmking, M., Juday, G.P., Barber, V.A. and Zald, H.S., 2004. Recent climate warming forces contrasting growth responses of white spruce at treeline in Alaska through temperature thresholds. *Global Change Biology*, 10, 1724-1736.

Referee #2 (Remarks to the Author):

Referee #2 comments “This study gives evidence that boreal conifers have expanded their populations into new areas, which is very interesting and the evidence using both satellite and field data seems to be compelling.”

AUTHORS' RESPONSE We appreciate this comment as we have made an attempt to integrate remote sensing and field data to present the story clearly and simply.

Referee #2 comments “Satellite data are mainly used to scale field observations to larger areas, and while the method is nothing special, it seems to work for this case. The study uses WorldView images, both WorldView-1 and WorldView-2, to manually count specific tree species which are only visible in winter, as they clearly stand out from the snow. The only evidence on the applicability of this approach is Figure 1, but it is not clear to me why the shapes under the yellow ovals are trees but not those on the left side, I really cannot see a difference. It is also a bit unclear to me if the study area is the shaded area from the small map.”

AUTHORS' RESPONSE We welcome the forthright criticism of Figure 1 by Referee #2 and have responded by removing ambiguity from Main Text **Fig. 1**. In addition, we provide more maps and images in **ED Fig 1** and **SI 1** and **SI 4.1-4.2**.

With specific reference to the shadows cast by trees in **Fig. 1a** we have now marked the base of each measured shadow with a small white disk and duplicated their location in **Fig. 1b** to highlight the difficulty in interpreting conifers in even very high-resolution growing season imagery compared to snow-covered scenes.

In addition, we now provide **SI Figs. 5-10**, and **SI Figs. 12-13** that show examples of shadows identified as trees in various contexts where we address the concerns of Referees #2 and #3 whom we invite Referees #2 and #3 to review in the context of our methodology.

With respect to the study area (AOI, area of interest), we define it as the four Noatak basin watersheds with spruce colonists and the four adjacent Kobuk basin watersheds with established treelines. The AOI is now outlined as the union of the 8 watershed boundaries (4 Noatak + 4 Kobuk basin watersheds) on **Fig. 1a** and throughout the four panels of **ED Fig. 1**.

Referee #2 comments “Artificial intelligence may be used in future studies to automate the detection process over much larger areas, where a manual counting is not feasible. Figure 1 also shows a color image, which is cited as “DigitalGlobe Vivid”, please make sure this is fine to show concerning copyrights.”

AUTHORS' RESPONSE Referee #2 suggests using AI to identify shadows. This is an excellent idea, we agree, and as an aside we note that we applied this approach in a pilot project, but found the wall-to-wall digitizing in our particular project was faster and more accurate with a human technician, but only because using the AI approach is new to us and outside our expertise.

We also appreciate the call to appropriately credit the imagery and have responded by placing “© 2022 Maxar Technologies” on each display item using these images, a standard followed by Google Earth Pro and elsewhere and as directed by our point of contact with Maxar.

Referee #2 comments “Figure 1 b says “revealing five trees”, but I can only see four yellow ovals.”

AUTHORS' RESPONSE One of the previous ovals encircled two trees. We hope the new Main Text **Fig. 1** is found to be clear, informative, and unambiguous.

Referee #2 comments “The method on the remote sensing part may be slightly improved and more details on many trees were counted, how this was validated, etc may be provided.”

AUTHORS' RESPONSE Remote sensing methods are now expanded, primarily in **SI 4.1-4.3**, but also in the Methods section. We hope that Referee #2 will find these descriptions, explanations, and validations improved.

We provide details on how we digitized in the Methods sub-section **Patterns of Expansion: Digitizing spruce shadows**. Details include metadata on positional accuracy of imagery and the actual digitizing process, details that are developed in **SI 4.1**.

Details are integrated in the subsection *Digitizing spruce shadows*:

Lines 692-714: We used cloud-free, Maxar Digital Globe WorldView-1 and WorldView-2 (WV) satellite scenes of snow-covered landscapes (Fig. 1b, ED Table s1, SI Figs. 1-13) from three missions in early spring 2018, a near-record year for snow depth in NW Alaska. Ground sample distances of 0.47-0.5 m, an RMSE of 3.91-3.94 m, and off-nadir angles of 5-25°, with low sun-elevation angles of 18-27° (<https://evwhs.digitalglobe.com/myDigitalGlobe/login>) provided clear images from which to digitize the lengths of individual spruce shadows and identify their locations (SI 4.1). One technician (ST in acknowledgements) supervised in QAQC by RJD digitized 5,986 shadows (densities in ED Fig. 1b, locations in SI Fig. 1) on Google Earth Pro (GEP) using WV images as super-overlays. The technician identified and digitized shadows as line segments from tree base to shadow tip wall-to-wall across the imported image tiles. The super-overlays degraded the imagery somewhat making small tree shadows more difficult to distinguish from snowdrift, rock, or shrub shadows. We suspect many trees in the 2-3 m height class were missed (SI 4.1-4.3). These line segments, saved as .kml files, were imported into R (v. 4.1.1)⁵⁹ using the sf package⁶⁰ where the length of each line segment was calculated and coordinates of the shadow's base identified. The line segment lengths were used to estimate tree heights (SI 4.2-4.3) and the coordinates used in nearest-neighbor calculations and extractions of gridded data values (SI 1). We estimated snow depth at 2.5-3 m because geolocated trees measured as ≤ 2.5 m in the field (see below) did not appear on imagery (ED Fig. 7b). We observed some trees taller than 2.5 m with no visible shadows in imagery (SI 4), possibly buried in deeper snow or growing in shadows cast by terrain at the time of image capture. Thus, our estimates of adult populations may be underestimates although there were also errors of commission where shadows were mistakenly classified as spruce (see following subsection).

Details provided in Methods subsection **Patterns of Expansion: Digitizing and field validation** include accuracy assessments of tree identification using shadows, the likely positional accuracy of GNSS devices used in the field to match shadows with trees, how field located trees were matched with image

shadows, and other details that are developed as examples in **SI 4.2**, a supplement that also includes a partial confusion matrix and multiple figures (*c.f.*, **SI Figs. 5-13**) illustrating the method, its challenges, and how they were addressed.

The subsection *Digitizing and field validation* is on

Lines 717-760: To estimate identification accuracy (SI 4.3) among 1,971 digitized shadows used for population reconstruction (enclosed by red rectangles in SI Figs. 1-4), we visited 157 shadow locations first identified on imagery (8% of the 1,971) and located in the field with the built-in GNSS of late model Apple iPhones (models 12 Pro Max, 12 Pro, and second-generation SE) with positional accuracy in the open landscapes estimated at 3 m. Of these 157 locations, 11 shadows were cast by very tall willows (7%). Of the 146 shadows confirmed as trees, two were dead (1%), and one had a recently broken top with green foliage on the ground. We added the broken top length to the standing height measured with a laser range-finder. Trees that were collinear in the solar azimuth at image capture contributed to errors of omission. The tree standing to solar azimuth obscured others as overlapping shadows fell in line, generating both errors of omission and an overestimate of the height of the first tree in the series. Six trees shadowed in three instances by what we identified on imagery as single shadows fell in this category. An additional three trees were missed during digitizing, also going unnoticed during QAQC, and discovered in the field when matching shadows with trees. SI 4.2 provides details and a confusion matrix.

In summary, 157 trees were expected from digitized shadows and 155 were identified in the field. Applying the accuracy of count overall suggests 1,945 trees would better estimate the reconstructed population. Across the AOI, the total adult count of 5,988 shadows may represent 5,910 trees. Moreover, in so far as our estimates of ages based on tree heights are predictive, perhaps 2% of the "trees" in our reconstruction are not a single tree casting long shadows, but 2-3

younger, collinear trees. Thus, our estimate of past populations may be slightly biased to older trees, implying that the population growth rate may be slightly higher than estimated. However, the slightly fewer trees than shadows would suggest that the growth rate is lower. The relative size of these errors, however, appears minor, and we did not incorporate them in the analysis which seems to us robust and perhaps conservative in adult abundance estimates due to image degradation with GEP super-overlays and other errors of omission.

Returning from the field with individual tree data, the first author displayed digitized shadow points together with field points on GEP, visually matching each field point to the nearest shadow, conditional on relative congruence between shadow size and tree height. This required care in clumps of trees with varying heights (example in SI 4.2). The relative patterning of field points compared to shadows and the lengths of shadows compared to tree heights in these cases provided some measure of confidence in attribution.

We made field expeditions to six study areas within the extent of the WV imagery we used for digitizing, three within “Simulated population area” rectangle of ED Fig. 1a (red rectangle in SI Figs. 1-4) and three study areas farther east (ED Fig. 1c, SI Fig. 2). Among-study area variability was apparent in snow depth, terrain slope relative to the solar azimuth at the time of image capture, and in the solar elevation angle itself, due to the timing of image capture. The variability was identified, calculated, and applied based on geographic variability in heights of trees casting shadows and from the slope and intercept of a mixed model linear regression of field-measured height on digitized shadow length (SI 4.3).

Referee #2 comments “Was it a wall-to-wall mapping or were only the clear ones mapped?”

AUTHORS' RESPONSE The mapping was intended to be wall to wall, whereby the technician identified and digitized all purported spruce shadows as line segments from tree base to shadow tip across the imported image tiles. **SI 4.1** describes how the technician differentiated spruce shadows from other shadows. However, because of image degradation when geo-tiffs were converted to GEP super-overlays, smaller spruce shadows were overlooked as described in **SI 4.1** and shown in **SI Figs. 5-6**. However, the technician intended to digitize every visible shadow that looked like it was cast by a spruce and not by a rock, snow-drift, shrub, or cottonwood (*Populus balsamifera*) tree.

Lines 698-702: The technician identified and digitized shadows as line segments from tree base to shadow tip wall-to-wall across the imported image tiles. The super-overlays degraded the imagery somewhat making small tree shadows more difficult to distinguish from snowdrift, rock, or shrub shadows. We suspect many trees in the 2-3 m height class were missed (SI 4.1-4.3).

Referee #2 comments “Also, I was looking for a map illustrating the results from the satellite image based mapping, but i only found ED Figure 2, which seems to be a small subset of the studied area, but it’s also not clear if this comes from field work or satellite images.”

AUTHORS' RESPONSE We hope that Referee #2 finds our new maps more informative with less ambiguous captions and legends.

For example, **ED Fig 1b** shows a rasterized image giving number of digitized spruce shadows per square-kilometer overlain on the extent of WV imagery that was used to identify the shadows. **ED Fig 1c** shows the locations of adults and **ED Fig 1d** shows the locations of juveniles measured in the field. **ED Figure 2a** remains a small subset of the study area that was used for the population simulation and is labeled as such, showing both rasterized counts of shadows and locations of select trees aged by increment cores collected in the field as bullets.

In the Supplementary Information, there are several figures showing the satellite image-based mapping. **SI Fig. 1** shows the mapping relative to named

geographic features on USGS topographic maps of scale 1:250,000, with the winter scene WV imagery overlain on the topographic maps, and the shadow locations overlain on the imagery. **SI Fig. 4. SI Figs 5-10 and 12-13** also show details of the mapping mostly in comparison to field data.

Referee #2 comments “ I assume ED Figure 1 shows the counts from the field work?”

AUTHORS' RESPONSE The new **ED Figure 1** supports four panels. The first, panel **a**, shows a thematic overview map with the AOI as a gray polygon, a central black rectangle identified as “Simulated population area”, Noatak and Kobuk Rivers in blue, shaded relief of the Baird Mountains of the Brooks Range, the hydrologic divide between Noatak and Kobuk river basins, the established treeline as digitized by ref. 34 (Maher et al. (2021)) in olive green, a scale bar, a north arrow, and the coordinate axes.

ED Figure 1b shows the same extent and scale as the first panel but shares with it only the AOI polygon and established treelines as thematic elements. In addition, the map provides a rasterized density of shadows in the AOI and the extent of the WV winter scenes used to digitize shadows.

ED Figure 1c shares the same thematic elements as panel **b** shares with panel **a** and additionally provides locations of field measured adults.

ED Figure 1d shares the same thematic elements as panels **b** and **c** share with panel **a**, as well as providing locations of field measured juveniles.

Referee #3 (Remarks to the Author):

Referee #3 comments “The authors have identified a previously unknown population of young spruce trees in Alaska, well beyond the tree line and expanding rapidly, possibly at an unprecedented rate.

AUTHORS' RESPONSE This is indeed the message we hope to convey.

Referee #3 comments “I am not enough of an ecologist to say whether this is in fact so, but I think well-founded research that bears upon the issue of the mismatch between climate change velocity and species advance rates in the subarctic is important.

“These rates differ by 2-3 orders of magnitude, so that the distribution of northern vegetation is increasingly out of equilibrium with the climate, and the question of what form approaches towards equilibrium will take is of great interest.”

AUTHORS' RESPONSE This framing of importance and interest is well articulated by Referee #3. We would be grateful if Referee #3 would allow us to paraphrase their language as follows in the Discussion:

Lines 206-217: The 20th century proliferation of spruce described here represents a climate-driven invasion of Arctic tundra at >4 km decade⁻¹, possibly matching estimates for post-LGM white spruce migration out of glacial refugia². Previous population reconstructions^{4-10,12,14,16} of spruce at their range limit found rapid infilling during the latter half of the 20th century, but the incremental range expansions documented¹⁵ in most studies (<0.1 km decade⁻¹) are an order of magnitude less than here. Examples of vigorous growth with rapid range expansion address the mismatch between climate velocity and boreal species advance. The orders of magnitude¹⁵ difference between rates of isotherm movement and forest migration leave vegetation increasingly out of equilibrium with climate. How modern forests will expand in the face of changing climate is poorly understood and so studies identifying instances of rapid advance can suggest mechanisms that increase LDD in seedlings, facilitate recruitment of saplings, and increase reproduction in adults.

Referee #3 comments

“The authors conclude their main text by suggesting that coniferous trees may be on the verge of invading the subarctic tundra. Perhaps this is a step beyond what can be supported by the research described here, but I am quite excited by this work.”

AUTHORS' RESPONSE To show that spruce invasion of arctic tundra is a more widespread phenomenon, we have included our observations from across the length of the Brooks Range, between Canada and the Chukchi Sea where we have located smaller populations of white spruce that are >1 km beyond established treelines (median 2.7 km). These are primarily juvenile spruce that are distant from established treelines and thus represent earlier stages of the invasion process. The locations within the Wulik, Noatak, Kobuk and Koyukuk basins are shown in a new map in the main text: **Fig. 1a**.

In the Discussion we added these three paragraphs:

Lines 288-309: The spruce population in the AOI, while perhaps the largest, is not the only one to recently colonize a tundra basin in Arctic Alaska. Within the Wulik and uppermost Noatak basins, spruce have dispersed across the boreal-Arctic divide in the Endicott, Schwatka, and De Long Mountains (Fig. 1a), apparently during the last three decades. Located 4.8-25.5 km from established treelines, these four populations of a few (1-3 encountered per site), small (15-60 cm), young (17-32 y) individuals most likely represent the first colonists to arrive in their respective Arctic watersheds, advancing at a median speed of 4.9 km decade⁻¹.

Across Alaska's Brooks Range, spanning 20° of longitude (143°-163° W) and equivalent to 20% of the species east-west range (63°-163° W), we found small populations of one to >70 individuals vigorously growing 1-25 km beyond established treelines (median = 2.7 km, n = 37 watersheds; Fig. 1a). The fewest colonized watersheds (n = 4) were encountered in the eastern Brooks Range (141°-148.3°), where winter precipitation is least. The most colonist populations (n = 19) were found in the central range (148.3°-155.7° W) near the triple divide of Koyukuk, Kobuk, and Noatak basins and where winter precipitation appears to be increasing (ED Fig. 6).

In one instance, 150 km east of the AOI and 25 km from the triple divide, we resurveyed a 4 km² valley previously censused⁴⁶ for white spruce above and beyond the established treeline. There, spruce have increased by a factor of 12 over 43 years, doubling every 1.2 decades. Where previously only seven spruce, all under 1.1 m, had been located over a three-year period, we located 70 juveniles and 19 adults (including nine cone-bearing) in three days.

And in the Methods under the section **Regional Extent of Colonization**:

Lines 1035-1060: Range expansion requires a sequence of successful life history stages: dispersal by seedlings, establishment as saplings, and sexual reproduction by adults. Over 22° of latitude (141°-163°) of the Brooks Range, the lead author (RJD) led field crews in search of ongoing range expansion by juveniles <1m tall representing successful dispersal (“seedlings”), juveniles >1m tall representing successful establishment (“saplings”), and individuals >2.5m tall representing potential reproduction (“adults”). The unit of sample was the watershed. Where one or more white spruce individuals (“colonist populations”) were encountered >1 km beyond established treeline, we recorded the location, age classes, and presence of cones when possible. In watersheds of the uppermost Noatak basin and the Wulik basin we also recorded both total height of juveniles and height above ground of the fifth budscar from the top to estimate RGR and so age. We encountered three watersheds with tree island krummholz >1 km beyond treeline, but do not include these as colonist populations. Among the 37 watersheds in which we encountered colonist populations >1 km beyond established treelines, four watersheds were located between 141° and 148.3° W; 19 watersheds between 148.3° and 155.7° W; and 14 watersheds between 155.7°-163.3°. Locations of watersheds west of 150.5°W with colonists are shown in Fig. 1a.

In 2021, RJD led a field expedition to a small watershed in the Koyukuk basin (Arrigetch Creek, 67.439° N 154.090° W). The watershed had been purposefully surveyed⁴⁶ for juvenile white spruce above and beyond treeline during 1978-1980 when seven juveniles 11-112 cm tall (six seedlings <1 m, one sapling ≥1 m) were located and mapped. Our resurvey of upper Arrigetch Creek found 70 juveniles (52 seedlings, 18 saplings) and 19 adults. Near the mapped location of the two tallest juveniles in ref. 46, RJD found a cone-bearing adult, as well as an additional eight cone-bearing adults elsewhere in the watershed, for a total of nine trees with cones among 19 adults up to 8 m tall. Four decades earlier the tallest tree reported had been ca. 1.1 m.

Referee #3 comments “I see it as primarily phenomenological research (I am less interested in the attempts to link the results to specific environmental factors in the study region), with a strong component of methodological development, and I would like to see it published to encourage more studies like this to be undertaken.”

AUTHORS' RESPONSE We appreciate this comment and have added a form of encouragement inspired by the last statement in the Referee #3 comment above to the Discussion as new text:

Lines 322-323: We anticipate that our observations will inform the search elsewhere for boreal forest advance.

Referee #3 comments

“Technical issues

“I have focussed, as requested (and as fits my expertise), on the remote sensing aspect of this study. Here, the authors have used very high resolution satellite remote sensing data as a means to estimate the height, and hence age class, of trees, together with their locations. The process is in general well described in the manuscript and in the portfolio

summary. However, it does raise some questions for me. I think these will probably all have satisfactory answers, but here they are:”

AUTHORS’ RESPONSE We hope that Referee #3 reads the comments from Referee #2 and our responses, as Referee #2 also provided insightful comments on and useful constructive criticism of the descriptions of remote sensing Methods. We hope that our responses to Referee #2 also inform Referee #3.

Referee #3 comments “1. The ability to match trees measured in situ with those identified in the WV imagery depends on the georeferencing accuracy of both the imagery and the GPS used for survey, as well as the spatial distribution of trees. The authors don’t give many details and it would be good to know how confident they are the matches are valid. In fact they report in the ‘portfolio’ section that field locations were obtained using iphones of various kinds, so (to be pedantic) these will have made use of several other GNSS systems as well as GPS, and I would expect the accuracy to be something like 3 m unless within dense forest. Were the WV images acquired close to nadir (i.e. easier to create orthoimages) or far from it?”

AUTHORS’ RESPONSE Thank you for pointing out the positioning systems in place on our phones and for asking about the matching of shadows with field measured trees. We have replaced “GPS” with “GNSS” throughout. We have included more information and images in the portfolio supplement, especially **SI 4.1-4.3** addressing challenges and accuracy of both positioning and matching with more abbreviated content in the Methods section **Patterns of Expansion** and its two subsections *Digitizing spruce shadows* and *Digitizing and field validation*.

In particular, in the *Digitizing and field validation* subsection we added this text:

Lines 746-751: Returning from the field with individual tree data, the first author displayed digitized shadow points together with field points on GEP, visually matching each field point to the nearest shadow, conditional on relative congruence between shadow size and tree height. This required care in clumps of trees with varying heights (example in SI 4.2). The relative patterning of field

points compared to shadows and the lengths of shadows compared to tree heights in these cases provided some measure of confidence in attribution.

Referee #3 comments “2. Almost 6000 shadow lengths were digitised by five individuals. Did all five measure all the shadows, or did they do non-overlapping samples? If so, did they make any attempt to identify and if necessary correct for systematic differences between them?”

AUTHORS' RESPONSE We apologize for our lack of clarity using the pronoun “we.” We use only the work of a single individual (initials ST) who digitized the shadows following training by the first author, who also performed QAQC and so did digitize <100 shadows that were missed but are also included with the dataset. However, there was no systematic or quantitative estimate of congruence, but informal “spot-checks” suggested that shadows were within ± 2 m in length of ST and RJD, which when multiplied by the tangent of sun elevation angle (18°) recorded on metadata for the imagery used to reconstruct the historical populations equals about ± 0.7 m in height. We have inserted this weakly supported observation in **SI 4.1**.

In the **Methods** section **Patterns of Expansion** and subsection *Digitizing spruce shadows* we have added the following text.

Lines 697-703: One technician (ST in acknowledgements) supervised in QAQC by RJD digitized 5,986 shadows (densities in ED Fig. 1b, locations in SI Fig. 1) on Google Earth Pro (GEP) using WV images as super-overlays. The technician identified and digitized shadows as line segments from tree base to shadow tip wall-to-wall across the imported image tiles. The super-overlays degraded the imagery somewhat making small tree shadows more difficult to distinguish from snowdrift, rock, or shrub shadows. We suspect many trees in the 2-3 m height class were missed (SI 4.1-4.3).

Referee #3 comments “3. Height classes deduced from the shadow measurements were in some cases only 0.5 m in width. If the mean snow depth (which has to be added to the shadow-deduced height) differed by only this much from one part of the study area to another, this would introduce a systematic misclassification between locations. How confident are the authors that this is not an important effect?”

AUTHORS' RESPONSE Referee #3 makes an important point concerning variability in snow depth and consequently the ability to capture accurate heights using our simple approach. We have added a sensitivity analysis in **SI 4.8**. We had recognized this variability in snow depth by way of the variability in heights of trees casting or not casting shadows (**ED Fig. 7b**), as well as the inherent variability in the angle of the terrain slope with respect to the sun elevation angle. These two components of the winter landscape combine to add spatially autocorrelated variability that we attempted to capture in the linear mixed effects model used to predict tree height as a function of shadow length presented in **SI 4.3-4.4**.

We felt that using the linear mixed effects modeling approach with random slopes and intercepts would alleviate some of this spatially correlated variability, a technique often used in experimental ecology, particularly because we used the fixed effects as the means in multivariate Monte Carlo sampling for all the simulations. For each simulation we drew a random slope and random intercept from the multivariate distribution with fixed effects as means but with variability determined by the random effects covariance matrix. The random effects covariance matrix captures the variability and covariation between the slope and the intercepts across the six study areas where we matched measured heights to shadows.

In **SI 4.3-4.4** we develop this mixed effects approach and graphically and systematically compare the variability in intercepts as they reflect snow depth. Here we found that variability in snow depth within study areas as determined by the difference within study areas between the tallest trees casting no shadows and the shortest trees that did cast shadows ranged from less than 0.1 m to 1.3 m (**SI 4.3.1**). In addition, variability across study sites in minimum field-measured trees heights of trees that did cast shadows was 1.6 m and for the maximum height of trees that did not cast shadows was 1.5 m. Thus, as expected there was more between-site variability in snow depth (up to 1.6 m) than within-site (up to 1.3 m). These numbers use measured tree heights to compare with presence or absence of shadows.

SI 4.8 takes the variability of 1.5 m in snow depth *between-sites*, which we attempt to capture through the mixed effects modeling approach, and performs a sensitivity analysis by setting the widths of size classes of trees for aging to 1.3 m to capture the *within-site* variability. Running the 1,000 simulations for the

broader height-classes gave surprisingly similar results for doubling times as reported in the main text.

Besides the additional **SI 4** material, we added text in the Methods:

Lines 853-862: Height classes deduced from the shadow measurements were in some cases only 0.5 m in width. Because the mean snow depth (the intercept in the mixed model) differed by more than this from one part of the study area to another (BobWoods, GaiaHill, and BuffaloDrifts in ED Fig. 7b and SI 4.2-4.4), this approach may have introduced a systematic misclassification between locations. While applying a Monte Carlo model with coefficients drawn randomly using `mvrnorm()` function from package MASS in R with the random effects covariance matrix was meant to alleviate this (SI 4.5-4.6), we also ran the simulation with three uniform height classes of wider intervals (1.3 m widths as 3-4.3 m, 4.3-5.6 m, and 5.6-7 m). The resulting mean doubling time was unchanged, but the variability increased when tree heights were binned more broadly (SI 4.8).

And in the Results:

Lines 141-143: On average, the simulated populations doubled every decade (median 9.5 y, IQR 8.7-10.6 y, SI 4.6) from 1900 to 1980 (Fig. 3b), a result robust to increasing bin width when reducing height-classes to three (median 9.8 y, IQR 8.8-10.8 y, SI 4.8).

Referee #4 (Remarks to the Author):

Referee #4 comments ‘I have reviewed with great interest the manuscript by Dial et al. “Sufficient conditions for climate-driven range expansion of a boreal conifer”. Overall, I think this is an important, complete and well-written piece that shows undisputable

evidence that the forest is advancing, benefiting, under certain circumstances, from global warming.

‘Ecological processes, like seed dispersion and other population-level ones, have been proposed to explain this vegetation advance.

‘My only major concern with the present study is the general idiosyncratic scope of it. Is *Picea glauca* colonizing other environmentally similar spots in the boreal region?’

AUTHORS’ RESPONSE From the comments of Referee #4, it appears that we succeeded in characterizing the large population of white spruce expanding rapidly across a large river basin in NW Alaska’s Arctic. The question posed is an important one and we have added two paragraphs to address it. We admit that our experience is limited to Alaska’s boreal-Arctic interface, the 1000 km between the Canadian border and the Chukchi Sea.

Motivated by the direct question of Referee #4 and by a similar, but unstated question raised by Referee #3 we address this point in the text.

But as a personal aside, perhaps an anecdote, the first author would like to say to the Editor and Referees here, that he has been looking for and documenting these white spruce range expansions across the Brooks Range since 2018, since the start of the authors’ collaborative NSF project on treeline. In 2019 the first author began formalizing a protocol in the eastern Brooks Range while working between the Alaska Pipeline and Canadian Border whereby he and his field crews actively searched for and recorded white spruce beyond treelines. These searches became more widespread and focused during subsequent 100-200 km vegetation transects ground-truthing remote sensing data between the Alaska Pipeline and the Chukchi Sea in 2020 and 2021.

What struck the first author, who first visited the AOI described in this manuscript in August 2019, was that the environmental conditions of the AOI could be used to predict, in an informal and as yet non-quantitative way, where else to look for spruce.

These searches are described in Methods under the section **Regional Extent of Colonization**:

Lines 1035-1060: Range expansion requires a sequence of successful life history stages: dispersal by seedlings, establishment as saplings, and sexual reproduction by adults. Over 22° of latitude (141°-163°) of the Brooks Range, the lead author (RJD) led field crews in search of ongoing range expansion by juveniles <1m tall representing successful dispersal (“seedlings”), juveniles >1m tall representing successful establishment (“saplings”), and individuals >2.5m tall representing potential reproduction (“adults”). The unit of sample was the watershed. Where one or more white spruce individuals (“colonist populations”) were encountered >1 km beyond established treeline, we recorded the location, age classes, and presence of cones when possible. In watersheds of the uppermost Noatak basin and the Wulik basin we also recorded both total height of juveniles and height above ground of the fifth budscar down to estimate RGR and so age. We encountered three watersheds with tree island krummholz >1 km beyond treeline, but do not include these as colonist populations. Among the 37 watersheds in which we encountered colonist populations >1 km beyond established treelines, four watersheds were located between 141° and 148.3° W; 19 watersheds between 148.3° and 155.7° W; and 14 watersheds between 155.7°-163.3°. Locations of watersheds west of 150.5°W with colonists are shown in Fig. 1a.

In 2021, RJD led a field expedition to a small watershed in the Koyukuk basin (Arrigetch Creek, 67.439° N 154.090° W). The watershed had been purposefully surveyed⁴⁶ for juvenile white spruce above and beyond treeline during 1978-1980 when seven juveniles 11-112 cm tall (six seedlings <1 m, one sapling ≥1 m) were located and mapped. Our resurvey of upper Arrigetch Creek found 70 juveniles (52 seedlings, 18 saplings) and 19 adults. Near the mapped location of the two tallest juveniles in ref. 46, RJD found a cone-bearing adult, as well as an additional eight cone-bearing adults elsewhere in the watershed, for a total of

nine trees with cones among 19 adults up to 8 m tall. Four decades earlier the tallest tree reported had been ca. 1.1 m.

In the Discussion we added these three paragraphs:

Lines 288-309: The spruce population in the AOI, while perhaps the largest, is not the only one to recently colonize a tundra basin in Arctic Alaska. Within the Wulik and uppermost Noatak basins, spruce have dispersed across the boreal-Arctic divide in the Endicott, Schwatka, and De Long Mountains (Fig. 1a), apparently during the last three decades. Located 4.8-25.5 km from established treelines, these four populations of a few (1-3 encountered per site), small (15-60 cm), young (17-32 y) individuals most likely represent the first colonists to arrive in their respective Arctic watersheds, advancing at a median speed of 4.9 km decade⁻¹.

Across Alaska's Brooks Range, spanning 20° of longitude (143°-163° W) and equivalent to 20% of the species east-west range (63°-163° W), we found small populations of one to >70 individuals vigorously growing 1-25 km beyond established treelines (median = 2.7 km, n = 37 watersheds; Fig. 1a). The fewest colonized watersheds (n = 4) were encountered in the eastern Brooks Range (141°-148.3°), where winter precipitation is least. The most colonist populations (n = 19) were found in the central range (148.3°-155.7° W) near the triple divide of Koyukuk, Kobuk, and Noatak basins and where winter precipitation appears to be increasing (ED Fig. 6).

In one instance, 150 km east of the AOI and 25 km from the triple divide, we resurveyed a 4 km² valley previously censused⁴⁶ for white spruce above and beyond the established treeline. There, spruce have increased by a factor of 12 over 43 years, doubling every 1.2 decades. Where previously only seven spruce,

all under 1.1 m, had been located over a three-year period, we located 70 juveniles and 19 adults (including nine cone-bearing) in three days.

Referee #4 comments “Literature from Canada is missing in this respect; probably, there is no literature documenting similar phenomena, but this is argument is not included in the Discussion.”

AUTHORS’ RESPONSE Referee #4 has made an excellent suggestion to briefly review the literature in Canada. We have added a paragraph in the Discussion that does so in two sentences, and includes a sentence about the rapidly advancing mountain birch in NW Eurasia and one reporting a recent global result from a remote-sensing paper:

Lines 311-320: While rates of movement appear slower than those described here, Canadian studies report that white spruce at their northeastern range limit in maritime Labrador⁷ and near Hudson Bay¹⁰ have advanced during the 20th century. In contrast, remote sensing studies in the vast, continental area between Hudson Bay and the McKenzie River Delta indicate leading-edge disequilibrium¹⁴ and even forest retreat¹² during the last half of the 20th century. The most rapid modern expansion¹⁵ of an Arctic tree so far reported is mountain birch (*Betula pubescens*), a small deciduous broadleaf tree advancing across Fennoscandia in northwest Eurasia at rates similar to those we report for white spruce. Meanwhile, Arctic-wide greening trends from high-resolution satellite imagery during 1985-2019 are consistent with a more global boreal forest biome shift northward⁴⁷ as temperatures continue to rise.

Referee #4 comments “I think it is important to bring some invasion ecology theory into the pattern studied but more importantly would be to scale up inference and show the potential for other areas in the boreal region to sustain similar vegetation advances.

For example, in the abstract it says: ‘This species range expansion, cast in the context of invasion theory, informs forecast models of vegetation change with conditions driving

biome shift from tundra to forest' and here it is where I would have expected to have the actual forecast including those vegetation models.”

AUTHORS' RESPONSE Referee #4 expected a prediction map given our abstract. That was due to poor wording in the abstract. That sentence has been rewritten as:

Lines 25-27: Cast in the context of invasion theory, this species range expansion may better inform forecast models with environmental conditions that convert tundra to forest.

In effect, Referee #4 points out that invasion theory, a useful theoretical construct, does not substitute for a map showing where to look next.

The second author recently published a paper that made such a map based on conditions associated with established treelines in northern Alaska (ref. 34, Maher et al. 2021). We feel that a mapping effort using what we have found here is beyond the scope of the manuscript submitted. It is an excellent topic for a future treatment of these data.

In part, Main Text **Fig. 1a**, the map showing where we have found other colonist populations, is the result of an informal application of what we have learned in the AOI. In 2021 we searched for and found spruce in every valley but one where we looked and expected to find spruce.

Referee #4 comments “High soil nutrient availability is described as a factor that could explain the establishment of forest in the new location; do you think a higher soil nutrient availability is a consequence of global warming? If so, in which way?”

AUTHORS' RESPONSE Generally, yes, warming is expected to increase decomposition and nutrient availability to arctic plants (Nadelhoffer et al. 1991). However, complexities are expected and have been observed in some permafrost environments. For instance, warming sometimes leads to collapse of

soils perched on permafrost; those soils often become water-saturated and paradoxically colder with more limited microbial processing of organic matter. Recently discovered complexities can also enhance the warming-induced increase in soil nutrient availability. At treeline in the Brooks Range, warmer winter soils associated with deeper snowpacks have been shown to lead to late winter labile carbon starvation of soil microbes and a consequent increase in inorganic nitrogen availability to plants (Sullivan et al. 2020)

Nadelhoffer, K.J., Giblin, A.E., Shaver, G.R. and Laundre, J.A. 1991. Effects of temperature and substrate quality on element mineralization in six Arctic soils. *Ecology*, 72, 242-253.

Sullivan, P.F., Stokes, M.C., McMillan, C.K. and Weintraub, M.N., 2020. Labile carbon limits late winter microbial activity near Arctic treeline. *Nature Communications*, 11(1), pp.1-9.

Lines 263-268: Thermal insulation provided by a thick snowpack promotes overwinter activity by soil microbes that can increase soil nutrient availability during the subsequent growing-season, an effect first proposed for Arctic shrubs⁴³⁻⁴⁴ and one that also appears to play a role in spruce growth. This increase in winter soil warming, together with growing season warming is expected to increase nutrient availability, a known limiting factor for spruce seedlings in tundra³⁹.

Referee #4 comments “In general, my concern centers on how peculiar the environmental conditions of the colonized area are. In other words, how generalizable these results are?”

AUTHORS' RESPONSE Referee #4 returns to the primary concern, one expressed by Referee #3 as well, that this might be an idiosyncratic AOI. We have presented the AOI because it is dramatic, large in area and in number of trees. It can be measured and compared in a statistical sense to established treelines. It also shows no sign of slowing. However, the 37 other sites we have uncovered, while smaller, less dramatic, and much younger, represent what the Cutler River basin in the AOI looked like 100 years ago. These locations offer

areas to monitor. They indicate that *Picea glauca* seeds disperse successfully beyond established treelines.

In **Fig. 1a** and among other populations distant from established treelines, we document four instances of Brooks Range crossings by white spruce other than within the AOI: two passes in the Schwatka Mountains to the Kugrak and Ipnellivuk watersheds, and one each in the De Long to the Wulik basin and in the Endicott Mountains to the uppermost Noatak tributary that abuts the Koyukuk basin. One-hundred years from now, will the Kugrak, Ipnellivuk, Wulik, and uppermost Noatak valleys will look like the Cutler does now, while the Cutler supports closed forests? We think so.

Reviewer Reports on the First Revision:

Referees' comments:

Referee #1 (Remarks to the Author):

I reviewed the prior version of this submission and mainly asked questions about the tree-ring work. I'm extremely impressed by the detailed and well-explained response provided by the authors — it's really one of the best I've ever received as a reviewer — and I am very grateful for the work they've done to engage with my questions and those from the other reviewers.

The authors make a fair point about the growth forms exhibited by these trees and I appreciate the concise description added to the Methods section.

The extended tree-ring sequences shown in Figure 3d address my earlier question about the long-term context of this recent growth acceleration. And the stacked scan added as ED Fig 4 is a wonderfully concrete illustration of the organism under the data. That's a great addition and much better than the spaghetti plot by itself.

The authors have carried out exactly the test I requested for their climate-growth comparison, and so I am convinced their autoregressive modeling of the tree-ring series has not created a spurious result.

The new caveat about the broader spatial representation of these results (in Lines 239-244) are very nice, thank you. I might suggest a revision to the wording to avoid some possible ambiguity — other tree-ring chronologies were collected one or two decades ago. And I think a parenthetical comment might be helpful to clarify that the end year of the Wilmking collections likewise stop almost two decades ago.

Finally, although it did not relate to my own questions, I enjoyed learning more about the context behind the study area as described in the response to Reviewer #4 and am glad those additions have been made to the Methods section.

Referee #2 (Remarks to the Author):

The authors have responded carefully to all comments from the reviewers. I have only one comment left concerning Figure 1:

I'm not sure if the white points are the perfect way to illustrate the presence of the tree, because it completely hides the object of interest, which may be irritating in the true color summer image. I leave it to the authors if they would like to change this or prefer to leave it as it is.

Referee #3 (Remarks to the Author):

Thank you for your extremely careful and very detailed response to the comments made by all three referees. I am, in brief, satisfied with those that relate to my own comments. As you have noted, some of the comments from referee 2 overlap with some of mine, and I have considered those too. (I did also read your response to referee 1, of course.)

You ask whether I would allow you to paraphrase some of my remarks in framing what is now lines 206-217. Yes, certainly, and I am happy with the proposed wording. I am also very pleased to read the additional text in lines 288-309, responding to my enthusiasm to see whether there was any more evidence to suggest that conifers are on the verge of invading tundra. I believe all my technical queries regarding the remote sensing methods have been well addressed (the new figures are very helpful, thank you). Where I think I was least confident was in the effect of snow depth variations on the relationship between shadow length and tree height, and I find the sensitivity analysis described in S14 and methods 852-861 persuasive on this point.

I have two very small comments to make now. First, I think 'QAQC' might need spelling out on first use. Second, perhaps it would be worth a comment about the slight degrading of imagery as a result of using the GEP super-overlays. There would be other ways (perhaps not in GEP, and perhaps not remotely as convenient) of capturing the digitisation data without degrading image quality, so I suppose this is something of a tradeoff.

Referee #4 (Remarks to the Author):

I have read again this manuscript and I am impressed, first, by the work the authors have done to reply to every comment brought by the reviewers (including mine), and second, by the study itself. I am now convinced this study makes a point of great relevance to the research of treelines and how this life form is responding in demography to global warming in northern North America.

I only have one further comment to make. Much has been discussed about the climate drivers controlling treeline formation (the limit of trees distribution at landscape scale) in the last decades, and mounting evidence worldwide points to the mean growing-season temperature as the one variable tightly correlated with treeline elevation, being this around 6.5 C (Körner's multiple papers). Thus, it results feeble to read in L256-7 that a 10C isotherm is the one bounding (boreal, only?) treeline, referencing an old technical report (nothing personally against it), while there are multiple more contemporaneous articles dealing specifically with the temperature threshold for treeline formation. I would expect some brief discussion about how temperature modules/drives treeline worldwide and how boreal treelines depart or not from this global pattern. Again, my point here (and before) is to scale your results up to a more global perspective, even if this is "just" verbally.

Author Rebuttals to First Revision:

Referee #1:

"I might suggest a revision to the wording to avoid some possible ambiguity — other tree-ring chronologies were collected one or two decades ago. And I think a parenthetical comment might be helpful to clarify that the end year of the Wilmking collections likewise stop almost two decades ago."

Referee #1 asks to change wording concerning previously collected tree-ring chronologies in the Brooks Range, with attention to the Wilmking collections.

On lines 160-162 we write:

"This is the highest proportion of adults responding positively to temperature recorded among treeline sites in the Brooks Range³⁴⁻³⁵. However, the spatially comprehensive Brooks Range chronologies presented in Wilmking et al.³⁴ and Wilmking and Juday³⁵ end with the 20th century, leaving uncertain how widespread recent rapid radial growth might be."

Referee #2:

"I'm not sure if the white points are the perfect way to illustrate the presence of the tree, because it completely hides the object of interest, which may be irritating in the true color summer image. I leave it to the authors if they would like to change this or prefer to leave it as it is."

Referee 2 suggests that leaving the actual trees visible in the imagery would be informative as they were hidden behind white bullets in Figure 1. We have replaced the white bullets with white circles in Figures 1b and 1c.

Referee #3:

"I have two very small comments to make now. First, I think 'QAQC' might need spelling out on first use."

Referee 3 asks to define "QAQC" at first mention. We have done so at Lines 624-628:

“One technician (ST in acknowledgements), supervised in quality assurance and quality control (QAQC) by RJD, digitized 5,986 shadows (densities in ED Fig. 1b, locations in SI Fig. 1) on Google Earth Pro (GEP) using WV images as super-overlays.”

“Second, perhaps it would be worth a comment about the slight degrading of imagery as a result of using the GEP super-overlays. There would be other ways (perhaps not in GEP, and perhaps not remotely as convenient) of capturing the digitisation data without degrading image quality, so I suppose this is something of a tradeoff.”

Referee 2 suggests a comment regarding the trade-offs in using GEP with super-overlays in image analysis. This is an important point and one that we were honestly unaware of until the final stages of this manuscript preparation when we were preparing Figure 1b in response to the first round of reviews and discovered how much sharper the original image was before conversion to the super-overlay. We have written on Lines 668-674:

“The relative size of these errors appears minor and we did not incorporate them into the analysis, which seems to us robust and perhaps conservative in adult abundance estimates due to image degradation with GEP super-overlays and other errors of omission. This study would have benefited from less image degradation using dedicated GIS or image software. However, the low-cost, simplicity and convenience of GEP was appealing for the large-scale digitizing.

Referee #4

I only have one further comment to make. Much has been discussed about the climate drivers controlling treeline formation (the limit of trees distribution at landscape scale) in the last decades, and mounting evidence worldwide points to the mean growing-season temperature as the one variable tightly correlated with treeline elevation, being this around 6.5 C (Körner's multiple papers). Thus, it results feeble to read in L256-7 that a 10C isotherm is the one bounding (boreal, only?) treeline, referencing an old technical report (nothing personally against it), while there are multiple more contemporaneous articles dealing specifically with the temperature threshold for treeline formation. I would expect some brief discussion about how temperature modules/drives treeline worldwide and how boreal treelines depart or not from this global pattern. Again, my point here (and before) is to scale your results up to a more global perspective, even if this is "just" verbally.

Referee 4 asks to put the study and worldwide response of treelines to warming in the context of Körner's hypothesis of direct control of physiological processes by temperature.

As the lead author, I am at fault for not previously citing any of Körner's extensive body of work. This oversight was due to our focus on Canadian and Alaskan data papers.

We have removed three citations in this most recent version: two omitted to bring the reference list down to 50, and a third replaced by Körner's recent, comprehensive review of treeline papers (Ref. 16: Körner, C., 2021. The cold range limit of trees. *Trends in Ecology and Evolution*, 36, 979-989). We chose this paper, rather than one of his seminal papers, because it is a well-written and insightful piece that places many concepts he has developed over the decades in a single place.

Ref. 16 is cited in several places including the introductory **Summary** and again in the last sentence of **Results with Discussion** at lines 255-258. We hope this is acceptable.

“Over time, warming will support higher latitude tree establishment, growth, reproduction, and dispersal, both directly¹⁶ and indirectly³⁴⁻³⁷, particularly with southerly winds, greater snowfall, and increases in nutrient availability, all warming-induced.”